# Model structures amplify uncertainty in predicted soil carbon responses to climate change

Zheng Shi [1,2], Sean Crowell[2], Yiqi Luo[3,4] & Berrien Moore III[2]

Large model uncertainty in projected future soil carbon (C) dynamics has been well documented. However, our understanding of the sources of this uncertainty is limited. Here we quantify the uncertainties arising from model parameters, structures and their interactions, and how those uncertainties propagate through different models to projections of future soil carbon stocks. Both the vertically resolved model and the microbial explicit model project much greater uncertainties to climate change than the conventional soil C model, with both positive and negative C-climate feedbacks, whereas the conventional model consistently predicts positive soil C-climate feedback. Our findings suggest that diverse model structures are necessary to increase confidence in soil C projection. However, the larger uncertainty in the complex models also suggests that we need to strike a balance between model complexity and the need to include diverse model structures in order to forecast soil C dynamics with high confidence and low uncertainty.

[1] Co-Innovation Center for Sustainable Forestry in Southern China, College of Biology and the Environment, Nanjing Forestry University, 210037 Nanjing, China. [2] School of Meteorology, University of Oklahoma, Norman, OK 73019, USA. [3] Center for Ecosystem Science and Society, Northern Arizona University, Flagstaff, AZ 86011, USA. [4] Department for Earth System Science, Tsinghua University, 10084 Beijing, China. Correspondence and requests for materials should be addressed to Z.S. (email: zheng_shi_ecology@outlook.com) or to S.C. (email: scrowell@ou.edu)

Human activities such as fossil fuel combustion and land use change are the dominant drivers of the fast increase in atmospheric $CO_2$ concentration[1,2]. The increase in atmospheric $CO_2$ concentration has been altering the climate system through additional radiative forcing[1]. Carbon-climate feedback is a major mechanism for regulating climate change. For example, terrestrial ecosystems can uptake about 1/3 fossil-fuel $CO_2$ emissions[3] and thus have the potential to slow down climate warming. Soils contain the largest carbon (C) stock in terrestrial ecosystems, twice as large as the content of the atmospheric C pool[4]. Therefore, a slight loss in the soil C stock to climate change may cause substantially positive feedback to the atmospheric $CO_2$, which could further warm the climate system. It is therefore essential to determine the sign and strength of such soil C-climate feedback.

Global land C models are critical tools for quantifying the response of soil C to climate change[5,6]. Large uncertainty in the predictions of soil C has been well documented among these models. Several model inter-comparisons (e.g., CMIP[7] and MsTMIP[8]) have demonstrated that global land C models vary considerably in their estimates of the global soil C stock for the contemporary period[5,9] and for the future greenhouse gas emission scenarios[6]. For example, Todd-Brown et al.[5,6] reported large differences in estimated contemporary global soil C stocks, ranging from 510 to 3040 Pg C projected by 11 Earth system models, and in projected change over 21st century ranging from a loss of 72 Pg C to a gain of 253 Pg C under the worst-case greenhouse gas emission scenario; Tian et al.[9] reported substantial differences in estimated contemporary global soil C stocks ranging from 425 Pg C to 2111 Pg C by ten terrestrial biosphere models in the Multi-scale Synthesis and Terrestrial Model Intercomparison Project (MsTMIP).

The global land C models differ in model structure, parameter values, and initial conditions, each of which may contribute substantially to the overall uncertainty across models. Past studies have shown that different model structures can generate different soil C projections[10–12]; initial conditions positively correlate with projected soil C content[5,13]; and classical parameterization causes large uncertainty in projected changes in soil C[14,15]. Model structure may determine the range of model projection and meantime the choice of parameter values for a given model structure defines the quantitative accuracy relative to observations. Therefore, parameterization is likely to interact with model structure and even initial conditions to impact the model projections. These considerations imply that diverse soil C decomposition models are needed to increase projection confidence[16]. Alternative structures to the conventional Century-type models include microbial models that simulate decomposition processes with explicit microbial traits as well as models that simulate interactions between soil layers at different depths[16–19]. Exploring uncertainty generated by these model structures and parameterization is critical for global land C modeling, but little effort has been dedicated to addressing it, especially at a global scale due mostly to computational cost.

In this work, we utilized a Markov Chain Monte Carlo (MCMC) technique to sample the parameter space for three different modeling frameworks in order to produce a calibrated ensemble of parameter values weighted by agreement with soil C observations. We chose three representative global soil C decomposition models with different structures (Fig. 1), among which were conventional Century-type model, a vertically resolved soil C model with explicit soil depth embedded in CLM 4.5[11] (CLM 4.5bgc) and a microbial model (the MIcrobial-MIneral Carbon Stabilization: MIMICS[20]). A set of projections driven by the Representative Concentration Pathway 8.5 (RCP 8.5) was carried out for all the three models with their posterior parameter ensembles to generate a distribution of predicted soil C, from which statistics such as the mean and uncertainty were estimated.

We hypothesized that uncertainty in projected soil C change by varying parameter values can be substantial: specifically CLM 4.5 and MIMICS would have larger projection uncertainty than the conventional Century-type model, especially the microbial model which is nonlinear and has a larger parameter space than the other two models; the two non-microbial models would predict

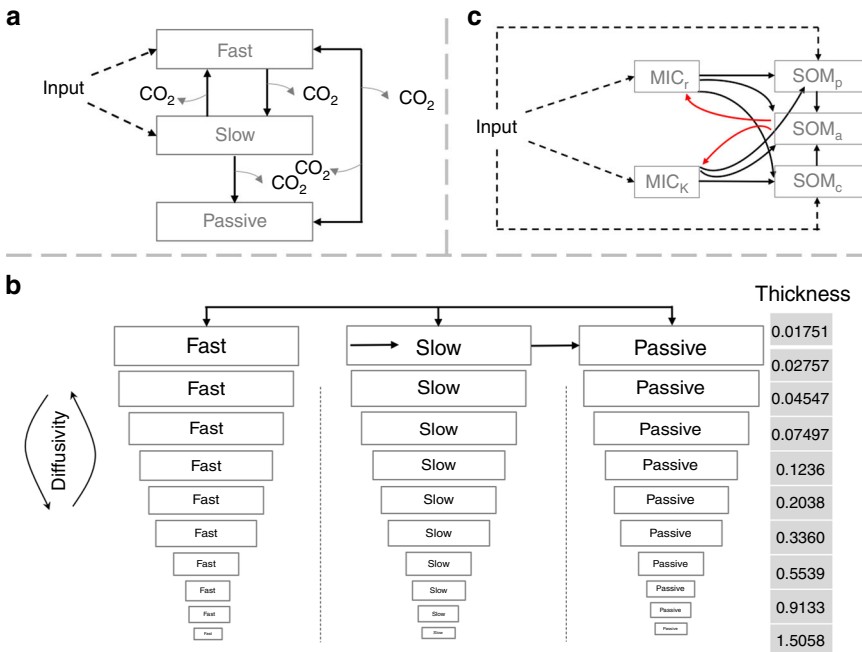

**Fig. 1** Soil carbon decomposition models with distinct structures. **a** Conventional Century-type model, **b** vertically resolved model in CLM 4.5bgc, and **c** a microbial explicit model, Microbial-Mineral Carbon Stabilization (MIMICS). MIC microbial biomass, SOM soil organic matter. Please refer to methods and Tables 1 and 2 and Supplementary Table 1 for parameterization. Unit of thickness of each soil layer is in meter

**Table 1 Descriptions of parameters in the conventional model**

| Short name | Description | Unit | LL | UL | Default | Mean ± SD | G-R |
|---|---|---|---|---|---|---|---|
| $t_1$ | Coefficient to calculate $f_{21}$ (intercept) | – | 0 | 1 | 0.85 | 0.12 ± 0.064 | 1.00 |
| $t_2$ | Coefficient to calculate $f_{21}$ (slope) | – | 0 | 1 | 0.68 | 0.048 ± 0.04 | 1.00 |
| $f_{31}$ | Fraction of C in fast soil C transferring to passive soil C | – | 0 | 0.01 | 0.005 | 0.006 ± 0.0025 | 1.00 |
| $f_{12}$ | Fraction of C in slow soil C transferring to fast soil C | – | 0.1 | 0.6 | 0.4185 | 0.47 ± 0.084 | 1.00 |
| $f_{32}$ | Fraction of C in slow soil C transferring to passive soil C | – | 0 | 0.05 | 0.0315 | 0.042 ± 0.0055 | 1.00 |
| $f_{13}$ | Fraction of C in passive soil C transferring to fast soil C | – | 0.3 | 0.7 | 0.45 | 0.50 ± 0.11 | 1.00 |
| $k_1$ | Turnover rate of C from fast soil C | $g\,C\,g\,C^{-1}\,yr^{-1}$ | 1 | 15 | 7.3 | 7.97 ± 3.72 | 1.00 |
| $k_2$ | Turnover rate of C from slow soil C | $g\,C\,g\,C^{-1}\,yr^{-1}$ | 0.1 | 0.5 | 0.2 | 0.28 ± 0.11 | 1.00 |
| $k_3$ | Turnover rate of C from passive soil C | $g\,C\,g\,C^{-1}\,yr^{-1}$ | 0.001 | 0.01 | 0.0045 | 0.0013 ± 0.00020 | 1.00 |
| $Q_{10}$ | Temperature sensitivity | – | 1 | 3 | 2 | 1.28 ± 0.054 | 1.00 |

*LL* lower limit, *UL* upper limit
Parameter names, ranges, units, default values, posterior mean, and standard deviation (SD), and G-R statistics in the conventional Century-type model

**Table 2 Descriptions of parameters in the vertically resolved model**

| Short name | Description | Unit | LL | UL | Default | Mean ± SD | G-R |
|---|---|---|---|---|---|---|---|
| $D_1$ | Diffusivity in non-permafrost regions | $m^2\,yr^{-1}$ | $0.3 \times 10^{-4}$ | $16 \times 10^{-4}$ | $1 \times 10^{-4}$ | $(1.4 \pm 0.81) \times 10^{-4}$ | 1.02 |
| $D_2$ | Diffusivity in permafrost regions | $m^2\,yr^{-1}$ | $0.3 \times 10^{-4}$ | $16 \times 10^{-4}$ | $4 \times 10^{-4}$ | $(9.3 \pm 4.0) \times 10^{-4}$ | 1.00 |
| $z_t$ | e-folding depth for depth scalar | m | 0 | 1 | 0.5 | 0.41 ± 0.05 | 1.00 |
| $t_1$ | Coefficient to calculate $f_{21}$ (intercept) | – | 0 | 1 | 0.85 | 0.28 ± 0.14 | 1.00 |
| $t_2$ | Coefficient to calculate $f_{21}$ (slope) | – | 0 | 1 | 0.68 | 0.089 ± 0.069 | 1.00 |
| $f_{31}$ | Fraction of C in fast soil C transferring to passive soil C | – | 0 | 0.01 | 0.005 | 0.0053 ± 0.0026 | 1.00 |
| $f_{12}$ | Fraction of C in slow soil C transferring to fast soil C | – | 0.1 | 0.6 | 0.4185 | 0.37 ± 0.13 | 1.00 |
| $f_{32}$ | Fraction of C in slow soil C transferring to passive soil C | – | 0 | 0.05 | 0.0315 | 0.031 ± 0.011 | 1.00 |
| $f_{13}$ | Fraction of C in passive soil C transferring to fast soil C | – | 0.3 | 0.7 | 0.45 | 0.50 ± 0.11 | 1.00 |
| $k_1$ | Turnover rate of C from fast soil C | $g\,C\,g\,C^{-1}\,yr^{-1}$ | 1 | 15 | 7.3 | 7.52 ± 3.76 | 1.00 |
| $k_2$ | Turnover rate of C from slow soil C | $g\,C\,g\,C^{-1}\,yr^{-1}$ | 0.1 | 0.5 | 0.2 | 0.22 ± 0.086 | 1.01 |
| $k_3$ | Turnover rate of C from passive soil C | $g\,C\,g\,C^{-1}\,yr^{-1}$ | 0.001 | 0.01 | 0.0045 | 0.0048 ± 0.0018 | 1.01 |
| $Q_{10}$ | Temperature sensitivity | – | 1 | 3 | 2 | 1.03 ± 0.024 | 1.00 |

*LL* lower limit, *UL* upper limit
Parameter names, ranges, units, default values, posterior mean, and standard deviation (SD), and G-R statistics in the vertically resolved model

decrease in soil C in response to climate change, but the microbial model could predict either negative or positive responses depending on parameter values; the projected soil C content would be correlated with initial conditions in the two non-microbial models but not in the microbial model. We found that CLM 4.5 and MIMICS with data-driven parameter values projected much greater uncertainties in soil C responses to climate change than the conventional Century-type soil C model, with both positive and negative C-climate feedbacks.

## Results

**Posterior distribution of model parameters.** Data-driven parameter ensembles were derived from assimilating re-gridded global soil C data in the Harmonized World Soil Database (HWSD[10]) and the Northern Circumpolar Soil Carbon Database (NCSCD[21]) (Supplementary Figs. 1, 2 and see Methods). We assumed that the soil C content in the HWSD and NCSCD was at steady state. Their probability distributions are log-normal (Supplementary Fig. 3). Applying Bayes' theorem, we used the MCMC technique to generate posterior ensemble parameter values within parameter boundaries (Tables 1 and 2, Supplementary Table 1). The probability inversion was effective in terms of constraining the targeted parameters in the three soil C decomposition models (Fig. 2, Supplementary Figs. 4, 5, 6). Specifically, coefficients (i.e., $t_1$ and $t_2$) for calculating $f_{21}$ (fraction of C in fast soil C transferring to slow

soil C), decay rate of passive soil C ($k_3$), and temperature sensitivity ($Q_{10}$) were well constrained in the conventional Century-type model (Fig. 2a and Supplementary Fig. 4); observed soil C vertical profiles further helped constrain the decay rate of slow soil C ($k_2$) in the vertically resolved model, particularly for the vertical diffusivity parameters ($D_1$ and $D_2$) and e-folding depth ($z_t$) (Fig. 2b, Supplementary Fig. 5). Interestingly, the posterior mean of $D_2$ was found to be larger than $D_1$, as diffusivity in permafrost soil was found to be faster than non-permafrost soil, which is mainly due to higher cryoturbation[11]. The $Q_{10}$ mean is 1.25 in the conventional model and 1.06 in the vertically resolved model, both of which are less than the default value (2), but close to empirical values[22]. The transfer coefficients ($f_{ij}$) were not well constrained in either of the two models.

In MIMICS, parameters related to uptake rate ($V_s$ and $V_i$) and desorption rate of physically protected soil C ($D_a$ and $D_b$), and proportion of litter input and the two microbial C pools ($f_m$, $f_s$, and $f_r$) were well constrained (Fig. 2c and Supplementary Fig. 6). None of the modifiers (e.g., $V_{mrc}$, $V_{mkc}$, $K_{mrc}$, and $K_{mkc}$) for calculating uptake rate and half saturation constant were well constrained.

**Steady states in soil carbon stock.** The spatial patterns of the estimated soil C content by the three models were comparable to the soil C database (Fig. 3a, c, e). The conventional model

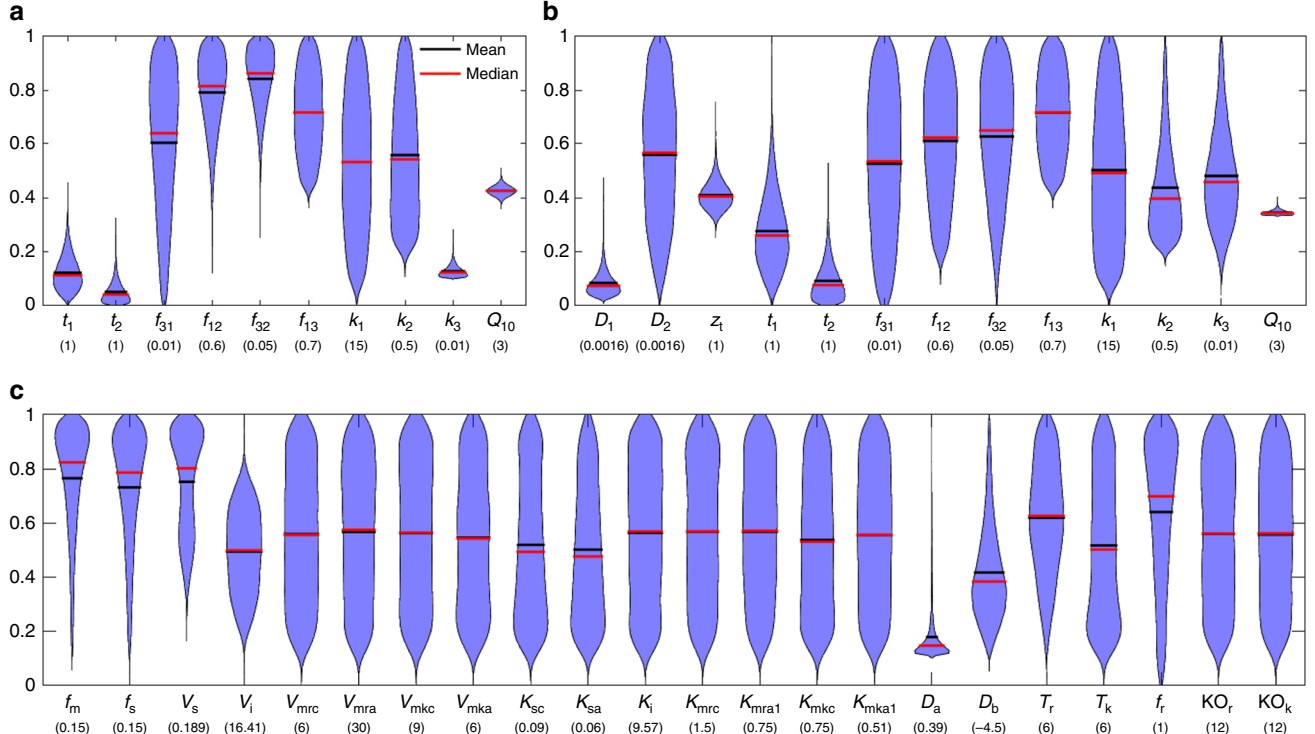

**Fig. 2** Violin plot of the accepted model parameter values in the three decomposition models. **a** The conventional model; **b** the vertically resolved model; **c** the microbial model (MIMICS). The narrower distributions of the parameters suggest that they are better constrained. For posterior distribution of each parameters, please refer to Supplementary Figs. 4, 5, and 6. For meanings of parameters, refer to Tables 1, 2, and Supplementary Table 1. Note that the values on the y-axis are normalized to the range of [0, 1] by the corresponding values in the brackets under each parameter. In the other words, the real values can be derived by multiplying the values in the brackets

significantly underestimated soil C in the high latitude (Fig. 3a) with large errors (Fig. 3b). However, the vertically resolved model generated relative small biases and errors in the high latitude (Fig. 3c, d) due to its vertical dynamics and explicit parameterization for permafrost soil. This finding highlights the advantage of adding soil layers to simulating high latitude soil C. The smaller soil C biases in the high latitude in MIMICS is due partly to the fact that observed soil C down to 1 m depth instead of 3 m was compared with the modeled soil C (Fig. 3e, f).

Positive biases compared to the observational data were widespread in the conventional and vertically resolved models in low latitude (Fig. 3a, c). In particular, the conventional Century-type model strongly overestimated soil C in tropical and near-coastal areas with large errors (Fig. 3a, b). The similar spatial biases between the two non-microbial models suggest the similarity in their structures. However, compared with the conventional model, the vertically resolved model had smaller errors, showing improvement by explicitly adding soil depth. In contrast to the two non-microbial models, the microbial model estimation showed small biases in the low latitude (Fig. 3e) with small errors (Fig. 3f). Overall, MIMICS estimated the best spatial fit to the observational data due possibly to having more parameters and explicit microbial dynamics[20,23,24]. As a result, CLM 4.5 and MIMICS generated smaller contemporary global total soil C relative to the observations, while the conventional Century-type model generated greater contemporary global total soil C in comparison to the observations (Supplementary Fig. 7).

**Uncertainties in soil carbon projections.** To illustrate the impact of parameter uncertainty on long-term soil C projection, we performed forward runs over 21st century with 1000 sets of

parameter values drawn from the posterior distribution for each model (Methods). The three models projected substantially different changes and trajectories in global total soil C over 21st century (Fig. 4). The conventional model projected consistent soil C loss with the least uncertainty (95% confidence interval: −71 to −17 Pg). Adding vertical resolution or microbial dynamics to the conventional model increased the projection uncertainty (95% confidence interval: −222 to 583 Pg C and −397 to 144 Pg C, respectively) as well as the sign of the soil C-climate feedback depending on parameters. The uncertainties in the vertically resolved model or MIMICS are more than 10 times larger than that in the conventional model. This interesting result shows that using more parameters and more explicit dynamics may lead to a larger prediction uncertainty due to feedbacks in the model dynamics, rather than less.

**Sensitivity to initial conditions and model parameters.** Uncertainties in projected soil C among models have been linked to the model initial conditions in previous research[5,13]. Our results show that the initial conditions ($S_i$) tightly correlated with the projected soil C in the two non-microbial models at global (Fig. 5; Supplementary Fig. 8a, c) and grid scale (Supplementary Fig. 9a, c), but did not correlate well with the changes in soil C in all the three models (Fig. 5; Supplementary Fig. 8b, d, f; Supplementary Fig. 9b, d, f) except for some low-latitude areas in the conventional model (Supplementary Fig. 9b). The microbial model's initial conditions were not correlated well with projected soil C at the global scale (Fig. 5 and Supplementary Fig. 8e) or grid scale (Supplementary Fig. 9e), except for significant correlations with predicted soil C at high latitudes (Supplementary Fig. 9e). The findings suggest that in general, uncertainty in the

**Fig. 3** Difference between estimated mean soil carbon and the soil carbon database. **a**, **b** are the difference and root mean squared error in the conventional model, respectively; **c**, **d** are the difference and root mean squared error in the vertically resolved model, respectively; **e**, **f** are the difference and root mean squared error in MIMICS, respectively

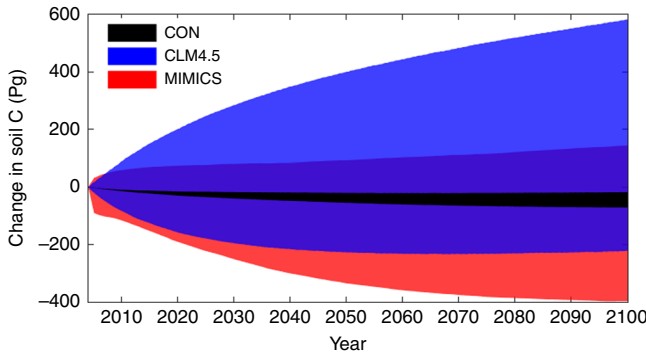

**Fig. 4** Changes in global total soil carbon under RCP 8.5. 1000 parameter sets were randomly sampled from the posterior distribution of parameters to generate the temporal trajectories in each model. The shaded area represents the 95% confidence interval generated by 1000 parameter sets. The time step is shown in year, from 2005 to 2100. CON in the legend is conventional Century-type model

initial conditions propagates through the simulation to the projection of future soil C, and this propagation is especially evident in the two non-microbial models.

Besides initial conditions, model parameters are also able to affect predicted soil C or C changes directly or indirectly through influencing initial conditions. In the conventional model, predicted soil C did not significantly correlate with any model parameters, but the initial conditions; changes in soil C were positively associated with $k_2$ (decay rate of slow soil C), but negatively with $k_3$ (turnover rate of passive soil C) and $Q_{10}$ (temperature sensitivity of soil C turnover) (Fig. 5). Besides positive correlation with initial conditions in the vertically resolved model, predicted soil C also positively associated with $D_1$ (diffusivity in non-permafrost soils) and $k_3$, but negatively with $k_2$; like predicted soil C, changes in soil C were positively associated with $D_1$ and $k_3$, but negative with $k_2$ (Fig. 5). Predicted soil C change weakly associated with $V_s$ (regression coefficient for calculating maximum reaction rate) and $D_a$ (coefficient for calculating desorption rate from physically protected soil C to available soil C); projected soil C content had no significantly linear relationships with any of the model parameters (Fig. 5).

Consistent with previous research, turnover rates often control soil C changes in the conventional model parameterizations[12,25]. In this study, $k_2$ is the key parameter for soil C dynamics in the two non-microbial models. However, the relationships between $k_2$ and soil C changes appear contradicted in the two models, positive in the conventional model but negative in the vertically resolved model. The possible reasoning is that conventional model mainly predicted C loss but vertically resolve model mainly predicted C gain; specifically, in the conventional model, more soil C would be transformed to the passive soil C with larger $k_2$ to minimize soil C loss; in the vertically resolved model, larger $k_2$ would cause more C loss to counteract soil C gain. In contrast, neither predicted soil C nor soil C changes were strongly

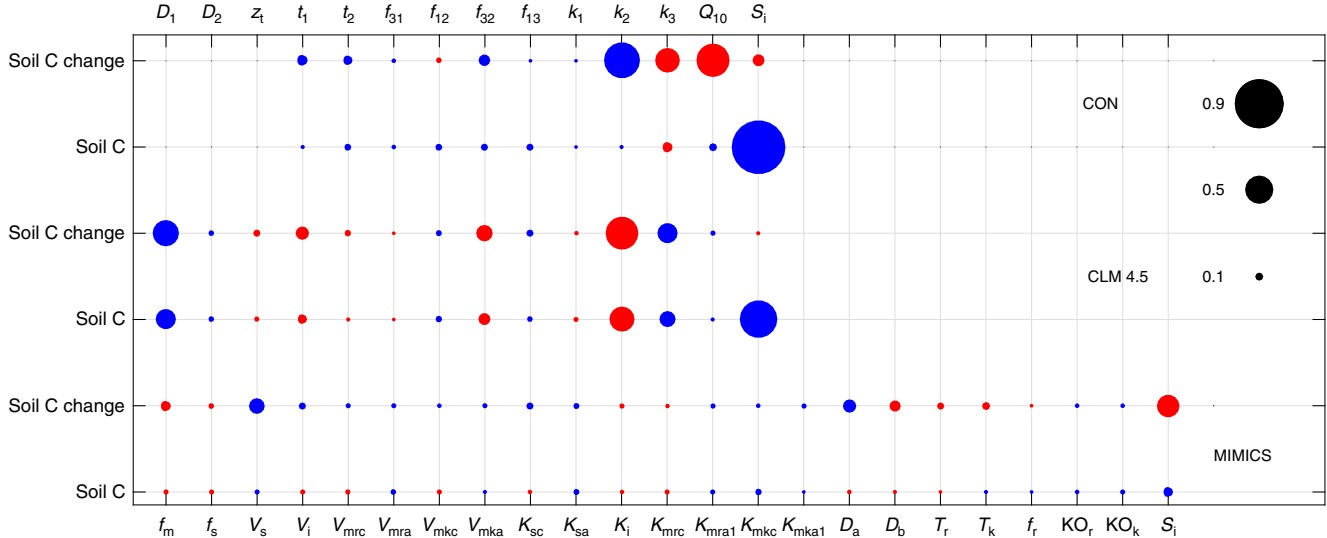

**Fig. 5** The relationships between soil C, model parameters, and initial conditions. The linear correlations between soil C or C changes, and initial conditions ($S_i$) and model parameters under RCP 8.5 in the three models. CON is the conventional Century-type model. Blue circles represent positive correlations and red circles represent negative correlations. The size of a circle is determined by the correlation coefficient between model parameters or initial conditions and soil C or C changes. The parameter labels on the top of the figure are for CON and CLM 4.5; the parameter labels at the bottom are for MIMICS. For the meanings of the parameters, please refer to Tables 1 and 2 and Supplementary Table 1

correlated with any single parameter in the microbial model, but weakly associated with parameters related to decomposition of physically and chemically protected soil C. This finding indicates the complexity of process dynamic and interactions in the microbial model.

## Discussion

Substantial uncertainties in soil C-climate feedback existed as a result of different model structures, parameter values, and initial conditions. The conventional Century-type model predicted consistently positive soil C-climate feedback with small uncertainty, which indicates effectiveness of data-driven projections. The consistently positive feedback suggests that the model structure determines the trajectory of soil C response to climate change for this family of models[19]. These projections are consistent with Hararuk et al.[14], who showed decreasing soil C under RCP 8.5 in a similar model to the one employed here, and are also within the predicted range in CMIP5 models[5].

We expected a similar, positive soil C-climate feedback in the vertically resolved model to that in the conventional model. Even though the vertically resolved model is parameterized with diffusivity of soil C across soil layers due to diffusion[11], the soil C decomposition in each layer has the same representation as the conventional model. Inclusion of the vertical dimension may not alter the fundamental behaviors of the model in terms of both steady-state estimation and long-term projection, as the soil C dynamics are still jointly determined by soil C influx and decay rate. In contrast to this hypothesis, adding soil layers to the conventional model allowed for both positive and negative feedbacks, due possibly to the smaller equilibrium soil C predicted by the model. The finding reveals that adding vertical resolution to the C decomposition parameterization can generate diverse responses of soil C to climate change. However, the large spread in projection and lack of constraint on transfer coefficients suggest more relevant data requirement to reduce model uncertainty.

The microbial model also predicted wider range of future soil C change and diverse trajectories with both negative and positive

feedbacks. This was somewhat expected due to the non-linearity of the C uptake processes by microbes[23,26,27]. The large spread in projections of the microbial model also suggests that reducing projection uncertainty requires more observations than are available at the present time in order to better constrain model parameters. Additional datasets are especially needed to tease apart multiple processes and further reduce the uncertainty. Results from this study highlight that data constraints may limit the ability of data assimilation to reduce uncertainty in more complicated model structures.

The large uncertainty in the vertically resolved model and MIMICS might be engendered by either high degrees of model freedom or complex model structures, or both. Many unconstrained parameters due to data limitation, especially in MIMICS, led to the large predicted uncertainty. We therefore anticipate substantial uncertainty reduction in these models once more global data are available to inform the models in the future. On the other hand, the complex model structures may also contribute to the large uncertainty. In a previous study, Hararuk et al.[14] reported small uncertainty in soil C prediction by a conventional model with 20 free parameters. The number of free parameters is comparable to that in MIMICS (22) and is greater than that in the vertically resolved model (13). Therefore, it is likely that the model structures of CLM4.5 and MIMICS at least in part increased the projected uncertainty. In addition, slightly larger uncertainty and fewer model parameters in the vertically resolved model than in the MIMICS also support that the greater uncertainty is likely caused by model structure, if not solely. In contrast to larger uncertainty in CLM 4.5 and MIMICS, both models generated better spatial comparison to soil C observations, which demonstrates model flexibility and encourages further exploration of explicit soil depth and microbial dynamics in soil C model parameterization.

The lack of constraint on the transfer coefficients in the two non-microbial models indicates that the two opposing mechanisms, transfer coefficient and turnover rate, require more informative data to be disentangled from one another. Substantial uncertainty in predicted soil C dynamics in MIMICS also suggests

that additional data are necessary to further constrain parameters. In particular, our results suggest that observations related to the modifiers of microbial maximum uptake rate and half saturation constant are of the most importance to reduce uncertainty. We acknowledge that besides transfer coefficients in the two non-microbial models and the modifiers in the microbial model, data related to other aspects of model processes may be also critical, but not revealed in our study due to model simplification. For example, data related to derivation of soil water scalar such as soil water content and water potential may be needed given that we simplified the calculation of water scalars by directly using the default parameters in the two non-microbial models and that there is under-representation of soil water impact in the microbial model[18].

Isotopic data in C processes show great potential to constrain these processes[11,28,29]. Indeed, [14]C soil profiles have been used to constrain transfer coefficients and turnover rates at multiple sites[28,30]; isotopic labeling to trace C pathways is another powerful tool to provide additional constraints to relevant processes, such as distinguishing root respiration from total soil respiration[31], sources of input[32], or proportion of different soil C pools[33]. Other additional data, such as soil respiration[34] and soil C incubation datasets[35,36] are equally valuable constraints. We therefore advocate using isotopic data and other datasets as complementary sources to better constrain model parameters and hence projections.

However, there are limitations with these additional datasets for model-data integration, especially at global scale. Most of these data are collected at small spatial scales, and hence may not provide a good global parameter constraint. Linking data at the micro-scale to the intermediate and large scales presents significant challenges for primarily two reasons: (1) lack of effective upscaling scheme may introduce additional uncertainty to the data; (2) data assimilation using global C models is difficult due to the fact that many parameters are global parameters with poorly understood regional variations with climate, vegetation, and edaphic properties. Leveraging these additional datasets as benchmarks instead of systematically assimilating them to constrain global model parameters may be the best use of these observations. Relaxing the global parameter assumption could be another option[15].

We caution the reader on several assumptions and simplifications in this study when interpreting our results.

First, we assumed the observed soil C dataset is at steady state due to its heterogeneity in time; second, we also assume the steady state in observed microbial biomass data used in the microbial model (MIMICS) due to the fast turnover rate of microbial processes. The steady-state assumption is convenient when the datasets are highly heterogeneous in time and space, but may introduce uncertainty in projection. Furthermore, these static estimates of soil C and microbial biomass C provide limited insights into the fate of soil C pools and potential microbial activity under climate change. However, it is a common practice so far to make best use of these datasets[10,12,14].

Another possible limitation relates to the two functional microbial groups in MIMICS. We did not estimate proportion of the microbial functional groups in each grid cell. Instead, we applied a global parameter (fr: proportion of r-selection microbial biomass) to calculate the r-selection microbial biomass, and the remainder is k-selection microbial biomass. We made this assumption due to the uncertainty in the spatial pattern of the relationship between the two microbial functional groups and climate, edaphic properties, and/or vegetation characteristics. Future research may focus on developing relationships between this parameter and climate, soil, and vegetation.

Lastly, we simplified the two non-microbial models in terms of their environmental modifiers, soil water scalar in particular, given data limitation and potential equifinality by the complex calculation of soil water scalar[37]. This simplification may underestimate the uncertainty in predictions by the two models; however, it is less likely for the conventional Century-type model to reach the similar magnitude of uncertainty in MIMICS even with the full representation of soil water scalar due to the large difference.

In summary, our results demonstrate the importance of model structure and parameterization in determining the predicted soil C response to climate change. CLM4.5 and MIMICS showed much greater uncertainty in projected soil C under RCP 8.5 and the conventional model consistently predicted strong positive C-climate feedback. The close correlations between initial conditions and projected soil C confirm that the projection of soil C is sensitive to initial conditions in the two non-microbial models, whereas the microbial model did not show any linear relationship between initial conditions and projected soil C.

To increase confidence in soil C projection, diverse model structures are necessary[16], given that CLM 4.5 and MIMICS outperformed the conventional model in terms of estimation in the spatial distribution of soil C. However, the larger uncertainty in the projection of soil C by the two models also suggests that we need to strike a balance between model complexity and the need to include diverse model structures in order to forecast soil C dynamics with high confidence and low uncertainty. In addition, reducing the uncertainties of the two models require more observations than are available at the present time. Overall, whether vertically resolved models and microbial models are better representations of mechanisms for soil C dynamics remain debatable. However, they represent updated knowledge and important alternate model structures to enhance confidence in prediction. Our findings suggest that the scientific community should include alternative model structures in future ensemble model predictions and comparisons to increase projection confidence.

## Methods

**Models**. We performed parameter estimation on three soil C decomposition models with different but representative structures (Fig. 1). The conventional Century-type model (Fig. 1a) has the same structure as soil C cascade embedded in the Community Land Model version 4.5 (CLM 4.5) without activating the depth-resolved parameterization; to incorporate the effects of depth-resolved dynamics, we used the model in the CLM 4.5[11,17] (CLM 4.5bgc; Fig. 1b); we used the MIMICS model[20] to explicitly simulate microbial soil decay processes (Fig. 1c). We briefly introduced the model parameterization in each model as follows.

The conventional model (Fig. 1a) represents soil C decomposition using three C pools and can be written in a matrix form as:

$$X'(t) = R + F \times \xi(t) \times K \times X(t). \tag{1}$$

**R** = [$R_1$, $R_2$, $R_3$] is the litter input to the three soil carbon pools (labile, slow, and passive soil C).

$$\mathbf{F} = \begin{bmatrix} -1 & f_{12} & f_{13} \\ f_{21} & -1 & 0 \\ f_{31} & f_{32} & -1 \end{bmatrix}$$

is a transfer matrix among soil C pools, where $f_{21}$ is derived using the equation $f_{21} = 1 - t - f_{31}$ where $t = t_1 - t_2 \times (1 - \text{sand}\%)$. The respiration coefficients can be derived by $1 - f_{i,j}$ for each carbon pool. **K** = [$K_1$, $K_2$, $K_3$] is baseline turnover rate of soil C pools. $\xi$ is the environmental modifier, and is a product of temperature scalar, soil moisture scalar, soil nitrogen scalar, and oxygen scalar. **X** = [$X_1$, $X_2$, $X_3$] is the soil C content in each of the three pools. To be consistent with the other two models, we only calculate temperature scalar using the equation $Q_{10}^{((T_{soil}-25)/10)}$, where $Q_{10}$ is temperature sensitivity of decomposition and $T_{soil}$ is soil temperature. The rest of the environmental scalars are average across the 10 soil layers from CLM 4.5. There are in total 10 global parameters in the conventional model. See Table 1 for detailed description of each parameter and its range and default value in the conventional model (i.e., CLM 4.5 without activating soil depth module).

For the vertically resolved model in CLM 4.5bgc, a matrix equation can be used to represent soil C dynamics among three pools within each soil layer over a vertical profile of 10 soil layers (totaling 30 pools, Fig. 1b). The three pools within one soil layer are labile, slow, and passive soil C. There are in total 10 soil layers with diffusivity among the layers. The matrix equation is:

$$X'(t) = R + F \times \xi(t) \times K \times X(t) + T_r \times X(t), \qquad (2)$$

where $R$ is litter inputs to soil C pools $\mathbf{R} = [R_{1,1}, R_{2,1} \ldots R_{m,n} \ldots R_{3,10}]^T$, $m$ is the soil C pool, ranging from 1–3, $n$ is the soil layer ranging from 1–10.

$$\mathbf{F} = \begin{bmatrix} F_1 & & & & \\ & \ldots & & & \\ & & F_L & & \\ & & & \ldots & \\ & & & & F_{10} \end{bmatrix}$$

is a block diagonal transfer matrix with dimension 30 by 30 (3 carbon pools per soil layer for 10 layers),

$$\mathbf{F_L} = \begin{bmatrix} -1 & f_{12} & f_{13} \\ f_{21} & -1 & 0 \\ f_{31} & f_{32} & -1 \end{bmatrix}$$

is a block matrix with L being the soil layers taking value from 1 to 10. The dimension of $F_L$ is 3 by 3 with element $f_{i,j}$, in which $i$ is a receiving pool, $j$ is a donating pool, and the blank in the matrix $\mathbf{F}$ are zeros. $\xi$ is environmental modifier, a product of temperature scalar, water scalar, depth scalar, and oxygen scalar as in Koven et al.[11]; $K$ is the baseline turnover rates for the soil C pools; $\mathbf{X}$ is C concentration for the 30 pools, $\mathbf{X} = [X_{1,1}, X_{2,1} \ldots X_{m,n} \ldots X_{3,10}]$, $m$ is the soil C pools, ranging from 1–3, $n$ is the soil layer ranging from 1–10;

$$\mathbf{Tr_{30 \times 30}} = \begin{bmatrix} Tr_{1,1} & Tr_{1,2} & & & & & & \\ Tr_{2,1} & Tr_{2,2} & \ldots & & & & & \\ & Tr_{3,2} & \ldots & Tr_{m-1,m} & & & & \\ & & \ldots & Tr_{m,m} & \ldots & & & \\ & & & Tr_{m+1,m} & \ldots & Tr_{8,9} & & \\ & & & & \ldots & Tr_{9,9} & Tr_{9,10} & \\ & & & & & Tr_{10,9} & Tr_{10,10} \end{bmatrix}$$

is a block tridiagonal matrix to represent C diffusivity between soil layers.

$$Tr_{m,m} = \begin{bmatrix} 0 & & \\ & tr_{2,2} & \\ & & tr_{3,3} \end{bmatrix}$$

is the fraction of a given C pool at a given soil layer being transferred into upper and lower soil layers,

$$Tr_{m-1,m} = \begin{bmatrix} 0 & & & & & \\ tr_{m-1,m} & & & & & \\ & tr_{m-1,m} & & & & \\ & & tr_{m-1,m} & & & \\ & & & tr_{m-1,m} & & \\ & & & & tr_{m-1,m} & \\ & & & & & tr_{m-1,m} \end{bmatrix}$$

is the received fraction of carbon transferred from lower soil layer m to layer $m-1$ and

$$Tr_{m+1,m} = \begin{bmatrix} 0 & & & & & \\ tr_{m+1,m} & & & & & \\ & tr_{m+1,m} & & & & \\ & & tr_{m+1,m} & & & \\ & & & tr_{m+1,m} & & \\ & & & & tr_{m+1,m} & \\ & & & & & tr_{m+1,m} \end{bmatrix}$$

is the received fraction of carbon transferred from upper soil layer, $m$ to layer $m+1$. Also, $tr_{m,m} = tr_{m-1,m} + tr_{m+1,m}$ for a given $m$. $\mathbf{Tr}$ can be approximated using the model parameter diffusivity ($D_1$ for non-permafrost diffusivity and $D_2$ for the permafrost diffusivity). Please see Patankar[38] Chapter 5.2 for details of the tridiagonal matrix calculation. Note that compared to the environmental modifier in the conventional model, there is an additional scalar, the depth scalar ($r_z$) which is computed with $r_z = \exp(-\frac{z}{z_\tau})$, where $z_\tau$ is the e-folding depth. To be consistent

with the other two models, we only calculate temperature scalar using the equation $Q_{10}^{((T_{soil}-25)/10)}$, where $Q_{10}$ is temperature sensitivity of decomposition and $T_{soil}$ is soil temperature. The rest of the environmental scalars are direct outputs from running CLM 4.5. Therefore, there are 13 global parameters in the vertically resolved model. See Table 2 for detailed description of each parameter and its range and default value in CLM 4.5.

The MIMICS model was developed in Wieder et al[20]. There are two soil microbial C pools ($MIC_r$ and $MIC_k$) and three soil C pools available soil C ($SOM_a$), physically protect C ($SOM_p$), and chemically recalcitrant C ($SOM_c$) (Fig. 1c). Michaelis–Menten equations are adopted to describe soil C uptake by soil microbes. The dynamics of the soil C can be represented by the following equations:

$$SOM'_p = R_{l-p} + R_{mic-p} - SOM_p \times D \qquad (3)$$

$$SOM'_c = R_{l-c} + R_{mic-c} - U_{c-k} - U_{c-r} \qquad (4)$$

$$SOM'_a = R_{mic-a} + U_{c-k} + U_{c-r} + SOM_p \times D - U_{a-k} - U_{a-r}, \qquad (5)$$

where $R_l$ is the input to soil C from litter ($R_{l-p} = f_m \times$ total_input and $R_{l-c} = f_s \times$ total_input) and $R_{mic}$ is the input to soil C from microbial decay, $D$ is the turnover rate for $SOM_p$, $U_{c-k}$ is the uptake of $SOM_c$ by k-selection microbes, $U_{c-r}$ is the uptake of $SOM_c$ by r-selection microbes, $U_{a-k}$ is the uptake of $SOM_a$ by k-selection microbes and $U_{a-r}$ is the uptake of $SOM_a$ by r-selection microbes. The uptake process takes the form of Michaelis–Menten equation, $MIC \times V_{max} \times SOM/(K_O \times K_m + SOM)$ where $MIC$ is the microbial biomass, $V_{max}$ is the maximum reaction rate, $SOM$ is the soil C content, $K_O$ is the modifier for oxidation of SOM, $K_m$ is the half saturation constant. For more model details, please see Wieder et al[20]. Slight modifications were made with MIMICS. Instead of using equations to estimate microbial turnover rates, we directly treat turnover rates as parameters as done by Hararuk et al.,[12]. In total, there are 22 global parameters (Supplementary Table 1). Since the range for most of the parameters is not well characterized in the literature, we prescribed the minimum of each parameter as the default values divided by three and the maximum as the default values multiplied by three. In addition to the soil C dynamics, the two microbial C pools are represented by the two equations in projection:

$$MIC'_r = R_{l-r} + U_{a-r} \times MGE1 - MIC_r \times \tau_r \qquad (6)$$

$$MIC'_k = R_{l-k} + U_{a-k} \times MGE1 - MIC_k \times \tau_k, \qquad (7)$$

where $R_{l-r}$ and $R_{l-k}$ are the input to r-selection and k-selection soil microbes.

$$R_{l-r} = (U_{m-r} + U_{s-r})/(U_{m-r} + U_{m-k} + U_{s-r} + U_{s-k}) \times (\text{Total\_input} - R_{l-p} - R_{l-c}) \qquad (8)$$

$$R_{l-r} = (U_{m-k} + U_{s-k})/(U_{m-r} + U_{m-k} + U_{s-r} + U_{s-k}) \times (\text{Total\_input} - R_{l-p} - R_{l-c}). \qquad (9)$$

$U_{m-r}$ and $U_{m-k}$ are the uptakes of metabolic litter by r-selection and k-selection microbes, respectively; and $U_{s-r}$ and $U_{s-k}$ are the uptakes of structural litter by r-selection and k-selection microbes, respectively; all the $U$'s are calculated with default parameters in MIMICS and litter from CLM 4.5 with the sole purpose of normalizing the input to r-selection and k-selection microbes. $U_{a-r}$ and $U_{a-k}$ are the uptakes of available soil C by r-selection and k-selection microbes, respectively. $MGE1$ is the microbial growth efficiency (MGE) for uptaking $SOM_a$. $\tau_r$ and $\tau_k$ are the turnover rates of r-selection and k-selection microbes.

Carbon use efficiency or MGE is a key parameter in microbial models[23,39]. However, we did not consider it as a parameter in our study due to that we used microbial biomass data[40] as an input to the MIMICS model. As a result, MGE is not involved in calculating soil C in MIMICS.

**Solutions for steady-state estimation.** To calculate the steady state in soil C content for each of the models, we made Eqs. (1), (2), (3), (4), and (5) equal zero and solved the equations for the state variables.

We derived the steady state ($\mathbf{X_{ss}}$) for the conventional model in the following form:

$$X_{ss} = -(F \times \xi(t) \times K)^{-1} \times R. \qquad (10)$$

The solution for the vertically resolved model is

$$X_{ss} = -(F \times \xi(t) \times K + Tr)^{-1} \times R. \qquad (11)$$

The solution for MIMICS is

$$SOM_{p,ss} = \left(R_{l-p} + R_{mic-p}\right)/D \tag{12}$$

$$SOM_{c,ss} = \left(-coeff\_f + \sqrt{coeff\_f^2 - 4 \times coeff\_e \times coeff\_g}\right)/(2 \times coeff\_e) \tag{13}$$

$$SOM_{a,ss} = \left(-coeff\_m + \sqrt{coeff\_m^2 - 4 \times coeff\_l \times coeff\_n}\right)/(2 \times coeff\_l), \tag{14}$$

where $R_{l-p}$ and $R_{mic-p}$ are the input to physically protected soil C from litter and microbial C, respectively; $D$ is the turnover rate of $SOM_p$.

$$coeff\_e = MIC_r \times V_{max\_r} + MIC_k \times V_{max\_k} - R \tag{15}$$

$$coeff\_f = MIC_r \times V_{max\_r} \times K_{o\_k} \times K_{m\_k} + K_{o\_r} \times K_{m\_r} \times MIC_k \times V_{max\_k} - (K_{o\_r} \times K_{m\_r} + K_{o\_k} \times K_{m\_k}) \times R \tag{16}$$

$$coeff\_g = -(K_{o\_r} \times K_{m\_r} + K_{o\_k} \times K_{m\_k}) \times R \tag{17}$$

$$coeff\_l = MIC_r \times V_{max\_r} + MIC_k \times V_{max\_k} - R \tag{18}$$

$$coeff\_m = MIC_r \times V_{max\_r} \times K_{m\_k} + K_{m\_r} \times MIC_k \times V_{max\_k} - (K_{m\_r} + K_{m\_k}) \times R \tag{19}$$

$$coeff\_n = -(K_{m\_r} \times K_{m\_k}) \times R, \tag{20}$$

where $R$ is the total input to respective soil carbon. For more details about these parameters, refer to Supplementary Table 1 and Wieder et al.,[20]

The steady state of total microbial biomass (sum of the r-selection and k-selection microbial biomass) was from a published database in microbial biomass[40] (https://daac.ornl.gov/SOILS/guides/Global_Microbial_Biomass_C_N_P.html). Some details are also provided in the data section below.

Note that all the litter inputs to soils in the three models are the same at grid level to avoid introducing bias into the results among models. The inputs were derived from running CLM 4.5 within 1850–2004. Mean annual input within 1850–2004 were used for steady-state calculation. Specifically, we used the Qian bias-corrected reanalysis dataset[41] to force the CLM 4.5 historical runs. Basically, for the model years 1850–1947, we cycle atmospheric forcing from the period 1948–1972, and use the corresponding atmospheric data for the years 1948–2004.

**Datasets.** The observations we used were re-gridded top soil organic carbon (0–30 cm, upper panel) and subsoil organic carbon (30–100 cm, lower panel) from HWSD [Food Agriculture Organization, 2012]. The native resolution (30 arc sec) in HWSD was re-gridded to the CLM grid (i.e., $1.25 \times 0.94°$) by Wieder et al.[10] (data source: https://daac.ornl.gov/SOILS/guides/HWSD.html).

Due to the possible underestimation of permafrost soil C in HWSD, we replaced it with the NCSCD (http://bolin.su.se/data/ncscd/netcdf.php). The NCSCD was developed to quantify the Northern Circumpolar permafrost soil C stocks down to 3 m[21]. There are four soil layers in this database, 0–30, 0–100, 100–200, and 200–300 cm. We regridded the NCSCD at 1° resolution to the CLM 4.5 grid, the same resolution as the HWSD and microbial biomass database.

We use NCAR Command Language (NCL) with Earth System Modeling Framework (ESMF) software. This software generated weights for data regridding between different resolutions. The conserve method was selected to insure the total amount of regional soil C to be conserved after regridding.

The microbial biomass C was used as the steady state in the MIMICS model due to the fast turnover of microbes, which is typically less than 1 year. A global parameter, fr (fraction of r-selection microbial biomass) was multiplied by the microbial biomass database to calculate the r-selection microbial biomass. The remainder is the k-selection microbial biomass. The gridded soil microbial biomass C was in 0.5° resolution and available here https://daac.ornl.gov/SOILS/guides/Global_Microbial_Biomass_C_N_P.html. The database was re-gridded to the CLM grid. The soil depth is down to 1 m[40].

Soil microbial biomass C data were re-gridded into $0.94° \times 1.25°$ resolution from $0.5° \times 0.5°$ resolution. We use NCL with ESMF software. This software generated weights for data regridding between different resolutions. The conserve method was selected to insure the total amount of regional soil C to be conserved after regridding.

**Model-data fusion.** We applied Bayes' theorem to estimate parameter values and associated uncertainties[25,42].

$$p(\theta|Z) = \frac{p(Z|\theta) \times p(\theta)}{p(Z)}, \tag{21}$$

where $p(\theta|Z)$ is the posterior distribution of the parameters $\theta$ given the observations Z. $p(Z|\theta)$ is the likelihood function for a parameter set calculated with the assumption that each parameter is independent from all other parameter and has log-normal distribution[5,43] (Supplementary Fig. 3) with a zero mean:

$$P(Z|\theta) \propto \exp\left\{-\sum \frac{[Z_i - \emptyset_i \times X]^2}{2\sigma_i^2}\right\}. \tag{22}$$

Here $Z_i$ is the logarithm of $i^{th}$ soil C observation in the observational database, $X$ are the logarithms of the carbon pools from the model, and $\emptyset$ is the mapping vector that maps the simulated carbon pools to observations. $X$ is derived by assuming the current soil status is at steady state. We were conservative in assigning errors to the soil C with $\sigma = 0.5 \times Z_i$. For the conventional model, we assimilated the data by aggregating all the soil layers together which is 0–100 cm in non-permafrost regions and 0–300 cm permafrost regions; for CLM 4.5, we assimilate data for 0–100 cm in HWSD for non-permafrost soils, 0–100, 100–200, and 200–300 cm in NCSCD for permafrost soils, independently. In contrast to soils elsewhere, permafrost regions contain a huge amount of carbon stock in deeper soil, which validates the use of deeper soil carbon data. For MIMICS, however, we assimilated the soil C data down to 100 cm only due to the explicit 1-m depth parameterization[20].

We assumed that the parameters are distributed uniformly within their prior ranges. Since most of the parameter range in MIMICS are unknown, we assumed the range of the distribution to be $[\theta_o/3, 3\theta_o]$, where $\theta_o$ is the default value. Posterior probability distributions of parameters were obtained using a Metropolis–Hastings (M–H) algorithm, a MCMC technique[44,45]. The detailed description of M–H algorithm can be found in Xu et al.[42].

In brief, the M–H algorithm consists of iterations of two steps: a proposing step and a moving step. In the proposing step, a new parameter set $\theta^{new}$ is proposed based on the previously accepted parameter set $\theta^{old}$ and a proposal distribution, which was uniform in our study:

$$\theta^{new} = \theta^{old} + r \times (\theta_{max} - \theta_{min})/D, \tag{23}$$

where $\theta_{max}$ and $\theta_{min}$ are the maximum and minimum values of parameters, $r$ is a random variable between $-0.5$ and $0.5$, and $D$ is used to control the proposing step size and was set to 5 as is Xu et al.[42]. In each moving step, $\theta^{new}$ was tested against the Metropolis criterion to examine if the new parameter set should be accepted or rejected. The first 2500 accepted samples were discarded (burn-in period) and the rest were used to generate posterior parameter distributions. In total, there are 50,000 accepted samples to construct the posterior distribution.

**Projection in soil carbon under RCP 8.5.** The soil C input and environmental modifiers (except the temperature scalar) used to drive the models were derived from running original CLM 4.5 under the worst-case greenhouse gas emission scenario, representative concentration pathway 8.5 (RCP 8.5). The atmospheric data were from Community Earth System Model output for the Representative Concentration Pathway 8.5 experiment, which were used to force CLM 4.5 for 2005–2100. We output the soil C input from litter and then used the inputs to drive our soil C models. The spatial and temporal changes in soil temperature, water content, and litter input to soil C within 2005–2100 were presented (Supplementary Figs. 10, 11).

We randomly sampled 1000 parameter sets out of the accepted posterior values. With each of the sampled parameter sets, we forced the vertically resolved model with C input and environmental modifiers obtained from CLM 4.5 model under RCP 8.5 from 2005–2100. For the conventional and microbial model, we derived total inputs and mean environmental modifiers of all the 10 soil layer to force the two models. For the two non-microbial models, we used a monthly time step. For the microbial model, daily time step was used due to its non-linear nature causing instability for longer time steps.

**Convergence of MCMC.** We used Gelman–Rubin (G–R) diagnostic method to determine convergence of MCMC simulations[46]. The idea of G–R test is that if the simulated Markov chain has reached convergence, the within-run variation within each chain should be roughly equal to the between-run variation among chains. Specifically, denoting each model parameter as $c_i$, the parameter samples from K ($K = 5$) parallel M–H runs of length N ($N = 10,000$), the between ($B_i$) and within-run ($W_i$) variances are defined as:

$$B_i = \frac{N}{K-1} \sum_{k=1}^{K} \left(\bar{c}_i^k - \bar{c}_i\right)^2 \tag{24}$$

$$W_i = \frac{1}{K(N-1)} \sum_{k=1}^{K} \sum_{n=1}^{N} \left(\bar{c}_i^{n,k} - \bar{c}_i^k\right)^2. \tag{25}$$

The G–R scale reduction statistics is given by:

$$GR_i = \sqrt{\frac{W_i(N-1)/N + B_i/N}{W_i}}. \qquad (26)$$

Once convergence is reached, $GR_i$ should approximately equal one.

**Data availability**. Data assimilation algorithms and parameter ensembles are available in the GitHub (https://github.com/zshi0609/Global-DA-Project). Large input datasets are available upon request to the correspondence authors.

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

## Acknowledgements

This research was supported by National Aeronautics and Space Administration (NASA) under OCO2 solicitation contract NNX15AJ37G. We would like to thank Chris Lu for the help in regridding soil carbon data. We appreciate the discussion with O. Hararuk, M. KJ, J. Jiang, P. Rayner, A. Schuh, and A. Chatterjee during development of the manuscript.

## Author contributions

Y.L., S.C., B.M., and Z.S. designed the study. Z.S. performed the anlayses. All authors contributed to the writing and discussions.

## Additional information

**Competing interests:** The authors declare no competing interests.

