## [Peer Review File · Nature Communications]

Reviewers' comments:

Reviewer #1 (Remarks to the Author):

I appreciate the revisions to the text and display items Shi and co-authors made to their paper. The additional information included in this longer format work make for a more thorough and interesting read. Aspects of the work, however, still could use some flushing out and discussion. Other aspects of the methodology, assumptions and approach can further be clarified.

Chiefly, I'd still like to see a more thoughtful interpretation of some of the relevant findings and a discussion of how they may 1) inform our understanding of soil biogeochemical processes or 2) be artifacts of the simplifications and assumptions made in this particular analysis.

Although I don't disagree with this conclusion "reducing the uncertainty of microbial models requires more observations than are available at the present time", I'm still concerned the analysis is cast too narrowly to honestly make the claim. For example, it's really the water scalar in CLM4.5 that shuts down decomposition in permafrost systems. I wonder if the low temperature sensitivity and large permafrost diffusion term (D2) reflects this omission? I'd suspect that if the authors really considered temperature vs. moisture sensitivities in the conventional models they also would have wider posterior estimates (fig 2) and future projections (Fig. 4).

Similarly, by omitting parameter estimate related to respiration coefficients (or MGE) it seems as though that another equifinality issues is conveniently sidestepped related to the K and F matrix in the first order models. Again, this generates several concerns related to the research priorities identified in the discussion / conclusion that focus on transfer coefficient and turnover rates (i.e., that are only related to temperature sensitivities in the present study).

I understand this reductionist approach is necessary to numerically constrain models. Given data limitations, however, are we at risk of being overly confident in getting the wrong answers for any multitude of the wrong reasons because of these simplifying assumptions?

Collectively, I guess these amount to philosophical questions, not particular scientific critiques of the manuscript. Perhaps a few lines can be added to the discussion along these lines, but this is a broader philosophical issue we can debate for years to come!

Technical concerns:

Line 55, I don't think the Todd-Brown papers look at 16 models

Line 105, please include posterior estimates in tables 1-3, as suggested in the text

Line 115, would the high D2 value be analogous to cryoturbation in permafrost soils?

Line 119. Several aspects of these findings are striking.

- The diffusion term (D2) seems critical for increasing permafrost C stocks in the vertically resolved model, and seems to largely be responsible for the correction of high latitude biases in the single layer model (Fig. 3). Isn't this just a fudge factor that allows for spatial 'corrections' in permafrost regions to better match the permafrost (NCSCD) data?
- The low Q10 values suggest that larger stocks are being stored at depth, and in slow/passive pools, which would have a lower climate sensitivity to climate change given their inherently lower

decay rates. This may ultimately be true, but several recent papers point to the potential vulnerability of permafrost C (e.g. Comane et al. 2017; Koven et al. 2017). * Note I realize the Koven paper is just out, but it suggests a yet lower e-folding depth is likely more appropriate for the same configuration of CLM4.5 and presents a quasi-independent dataset that can be used to evaluate SOM models.

- CLM4.5 was tuned up using the vertically resolved isotope data discussed in the paper (Koven et al. 2013). Do these fitted global parameter combinations maintain soil profiles that look anything like the observations Charlie used in his paper? Maybe this is beyond the scope of the paper, but do the spatial distributions of stocks in the vertically resolved simulation correspond to observed estimates (Fig. S3)?

Line 127, Fig S7. Are these total values for CLM4.5 for the top 100 cm soil, or the whole 3.5 m profile?

Line 162, I'm still unclear how or why the authors prescribed initial conditions (Fig. 4). I thought the purpose of the data assimilation that provides the bulk of the paper was to generate initial conditions that reflect observationally derived estimates? By proscribing initial conditions won't there be spurious trends in the transient soil C response?

Line 267, parametric and structural (process) uncertainty strike me as distinct issues. Yes, adding processes also increases parameters, but I feel distinguishing among these sources of uncertainty is important

Line 286 Part of the assumption in MIMICS is that environmental change (e.g. increased plant productivity) will increase microbial biomass pools (Wieder et al. 2015). Is microbial biomass allowed to fluctuate in the transient simulations, or is it held constant in the runs? Please clarify here (and? in the methods?).

Line 313, is this statement still true with the new analysis using the HWSD?

On an unrelated note, I wonder if the various parameter estimations actually afford enough SOM turnover to generate enough inorganic N to meet plant and heterotrophic N demand in the first-order models. The present approach assumes that modifications to the parameter estimates don't influence the nitrogen scalar applied in CLM4.5 (line 339). This is impossible to assess without looking at litter dynamics too (which generally immobilize N), but it's an interesting feedback that's not considered in the present analysis. I'm not suggesting it should be included in the analysis but was struck by this limitation of the approach (or opportunity to throw out parameter combinations that may make the N cycle unrealistic).

Just to clarify, CLMcn has a distinct structure and parameterization used in version 4 of the Community Land Model that includes 3 litter and 4 soil carbon pools. This structure is not applied here and should not be referred to in the manuscript (I think it's currently only used in the response to reviewers).

Methods & Display items:

Line 343. Why are environmental scalars being averaged over the 10 soil layers in CLM4.5? This covers a depth interval of 0-3.5 m, whereas soil C pools are only estimated for 0-1 m in HWSD and in MIMICS? (see also comments above related to the C stocks for vertically resolved model).

Line 524. I'm still not really clear on nuances of the future scenarios work. How did the authors transition from this data atmosphere to the modeled (RCP8.5 atmosphere)? Is it worth showing (or describing) changes in litter inputs, temperature, moisture, and their spatial distribution associated with these simulations in the SI? There are often abrupt step changes when moving from a data to model atmosphere. I'd assume such changes would cause more non-linearities in MIMICS. Thus, to what degree is the uncertainty communicated in Fig. 4 related to the switch between atmospheric

forgings?

Line 533, as above, why 10 soil layers?

Can another column be added to Tables 1-3 showing the mean (or median) estimates for posterior values for each parameters (and their uncertainty)? I like the information communicated in Fig. 2, but this additional data in the appropriate tables would be useful too.

Is vertical mixing (Fig 1b) the same as diffusivity (Table 2)? If so, can consistent terminology be used throughout on display items

Fig 1c: I'm still confused where the C fluxes go here. If microbial biomass is fixed based on the Xu dataset (line 443), but C inputs enter the MICr and MICK pools is the model effectively being tuned to get the 'right' microbial biomass pools? Without MGE, the only modifier on microbial biomass is the tau term, which is also critical for determining steady state SOM pools (Wang et al 2013)? To maintain these biomass stocks in the RCP scenario, tau would have to increase, and would necessarily build more SOM pools (without a feedback to increase decay rates via larger microbial biomass pools)? Am I understanding this correctly?

Table 3. I'm not sure what the following parameter refer to in the modified implementation of MIMICS, as litter stocks have been removed and chemically protected SOM doesn't flow into microbial biomass (Fig. 1) (Vmrc, Vmkc, Kmrc, Kmkc). As such, I wonder if these parameters should be removed from the analysis, or at least their description in the table should be clarified? Also, how are Kmrc, Kmkc different from KOr and KOK?

MIMICS makes explicit assumptions about the kinetics and turnover of r vs. K microbial communities (Wieder et al. refs in the paper). Do the parameter combinations used in the data assimilation maintain these relative differences (e.g. faster kinetics and turnover of copiotrophic communities)? If not, then differences between r and K physiology are meaningless, and distinctions made between their physiology and relative abundance should be removed from the analysis.

The colors used in the left panels of Fig 3 are ugly. The RMSE colors are nice, but they don't communicate the sign of the biases like the left panels do.

Fig 5, the whole text is organized with the single layer, multi-layer, and MIMICS model organization, should this figure follow the same layout (top to bottom)?

Fig 5, what is Si (right side of figure). It's never mentioned in tables 1-3 (as stated in the caption), but has large leverage in the first order models?

Fig. S3, doesn't the HWSO just cover 0-100 cm? Please clarify what's shown in these histograms

Commane R, Lindaas J, Benmergui J et al. (2017) Proc Natl Acad Sci U S A, 114, 5361-5366, doi:10.1073/pnas.1618567114.

Koven CD, Hugelius G, Lawrence DM, Wieder WR (2017) Nature Clim. Change, advance online publication doi:10.1038/nclimate3421.

Sincerely,
Will Wieder

Reviewer #2 (Remarks to the Author):

Dear authors

The uncertainty of the representation of soil carbon dynamics in Earth System Models is an important issue and this manuscript addresses the structural uncertainty with three models of increasing complexity.

I have carefully read your replies to previous reviews and find that this revised manuscript has addressed most of the concerns previously raised.

I am, however, concerned about the spread of the uncertainty in projections made with the MIMICS model. While I appreciate that this study presents results from offline models, an uptake of 2,000 Pg C would mean a doubling soil stocks over the course of one century... This should have been addressed as a main limitation of the study in the paragraph I. 280 onwards.

Also, the statement that "Our results imply that the scientific community should include microbial models in future ensemble model predictions and comparisons to increase projection confidence" contradicts the results. It feels that MIMICS is actually over-parameterized for the low amount of constraints used in the current study, something that is referred to as "model flexibility" (I 244).

These points of discussion are raised in sections starting on I 249 and I 280. Therefore, I don't think these points have to be addressed with new experiments for this paper, but perhaps more emphasis should be put on how realistic these simulations are, in the discussion.

I am not sure of the usefulness of the experiments based on "data-derived" initial conditions. These results are only mentioned once in the text (I 167) and, as the authors correctly mention, findings are anticipated as the data-derived initial conditions are most likely not the same as the models' own steady-state. Perhaps you could consider removing this experiment for the sake of simplicity.

Finally, it seems unrealistic for me to reach exactly 1 when calculating the G-R criterion across multiple chains. The number of chains used in the MCMC is not indicated, neither is the precision of the G-R criterion reported in Tables 1, 2 and 3. This needs clarification as, at the moment, it seems that only one chain has been used which would involve a risk of being trapped in a local optimum.

Hereafter are some minor suggestions for improvement:

I 43: "can uptake about 1/3 fossil-fuel CO₂ emissions",

I 105-106: here or under methods, please indicate how you use the constraint with the multi-layer model... is it summing up all C stocks in each layer? or do you assimilate data for 0-100, 100-200 and 200-300 cm independently?

I 153-157: please give a quantitative comparison of the uncertainty between the different model structures, e.g. how much larger is the uncertainty in the vertically-resolved / microbial model compared to the conventional model?

I 157-160: see my main point, how realistic is it for global soils to uptake 2,000 Pg C over the course of a century?

I 221-222: what about other models used in e.g. cmip5? I assume that our results fall well within the uncertainty of projections obtained by other conventional soil carbon models used in most ESMs.

I 305: I do not think the term C-climate feedback is adapted here. All experiments are made with prescribed atmospheric conditions. What your results indicate is a gain or loss of soil C.

I 348-350: what is the thickness of each layer?

I 468-472: these two sentences are redundant

I 527: where do the atmospheric data used to drive CLM come from?

Table 1: what is the precision of the G-R statistic reported here? It seems unrealistic to reach exactly 1 when calculating it across several chains. The number of chains is also not reported.

fig 2: many parameters in MIMICS are not constrained, is it a problem?

fig S3: please indicate that data for 100-200 and 200-300 cm layers is only available in NCSCD

fig S7: please add the uncertainty of the composite database on the figure

fig S8: please indicate that the correlation is calculated across the ensemble of 1,000 global simulations

fig S9: same as above

1 **Responses to Comments from Reviewers**

(Manuscript number: NCOMMS-17-20128-A)

**Dear Reviewers:**

We appreciate your insightful comments and suggestions. Below, we address the comments and
questions point-by-point. We have made changes in the text accordingly and highlighted them in
red. The major points are listed as follows:

**The soil depth in MIMICS:** Dr. Wieder mentioned that MIMICS simulates soil carbon (C)
down to 1 meter and reviewer #2 pointed out that some predictions in MIMICS are not realistic
(“*an uptake of 2,000 Pg C*”). Based on these comments, we changed the assimilated and
predicted soil C from 3-meter deep to 1-meter deep. With these changes, the predicted
uncertainty in MIMICS reduced substantially and all predictions seem reasonable. Please see
figure 4 in the revised manuscript. Now the uncertainty in MIMICS is even slightly smaller than
that in the vertically-resolved model, which further increases our confidence in advocating more
involvement of microbial models in scientific endeavor. Fortunately, the new result does not
change our conclusion (alternative model structures to the conventional model amplify predicted
uncertainty).

**Thoughtful discussion of the key findings:** Dr. Wieder suggested that “*more thoughtful*
*interpretation of some of the relevant findings and a discussion of how they may 1) inform our*
*understanding of soil biogeochemical processes or 2) be artifacts of the simplifications and*
*assumptions made in this particular analysis.*”. We have updated multiple places in the revised
manuscript to reflect the comments. Specifically, we discussed the possible under-estimation of
uncertainty due to not involving water scalar calculation in the data assimilation process in the
conventional model; we discussed whether the large uncertainty in the two alternative model
structures are caused by high degree of model freedom or model structure or both; we also toned
down the suggestion of collecting more data on transfer coefficients and turnover rates. Please
see responses to comment 2, 3, 4 and 5 for more details.

**Removal of data-derived initial condition:** Our description on the prescribed initial condition
was not clear to either of the reviewers. Dr. Wieder commented that “*I thought the purpose of the*
*data assimilation that provides the bulk of the paper was to generate initial conditions that*
*reflect observationally derived estimates?*”. Reviewer #2 also pointed out that “*I am not sure of*
*the usefulness of the experiments based on "data-derived" initial conditions. These results are*
*only mentioned once in the text (l 167) and, as the authors correctly mention, findings are*
*anticipated as the data-derived initial conditions are most likely not the same as the models' own*
*steady-state. Perhaps you could consider removing this experiment for the sake of simplicity.*”.
We agree with both reviewers and have removed this part for the sake of simplicity and clarity.
Please see figure 4 in the revised manuscript. Relevant information on prescribed initial
condition has also been removed. We further changed the title to reflect the removal.

**Convergence in posterior probability distribution:** Reviewer #2 was concerned with the G-R
criteria due mainly to the precision. We have added more information and updated the precision
with the accuracy of two decimal places. Please see table 1, 2 and 3 and our response to
comment 36.

Thank you very much for considering our revised manuscript. We hope you will find our
revision satisfactory.

Sincerely,
Zheng Shi
School of Meteorology
University of Oklahoma, OK, USA
Tel: +1 (405) 325-6519
Email: zheng_shi_ecology@outlook.com

**Response to Reviewer #1 Dr. William Wieder**

[Comment 1] I appreciate the revisions to the text and display items Shi and co-authors made to
their paper. The additional information included in this longer format work make for a more
thorough and interesting read. Aspects of the work, however, still could use some flushing out
and discussion. Other aspects of the methodology, assumptions and approach can further be
clarified.

**[Response]** We thank Dr. Wieder for the positive evaluation and appreciate these constructive
comments. Please see our responses below to each comment.

[Comment 2] Chiefly, I'd still like to see a more thoughtful interpretation of some of the relevant
findings and a discussion of how they may 1) inform our understanding of soil biogeochemical
processes or 2) be artifacts of the simplifications and assumptions made in this particular
analysis.

**[Response]** We appreciate these great comments. To more thoughtfully interpret and discuss our
key finding (model structures can amplify uncertainty in prediction), we have updated multiple
places. To summarize:

“The large uncertainty in the vertically-resolved model and MIMICS might be engendered by
either high degrees of model freedom or complex model structures, or both. Many unconstrained
parameters due to data limitation, especially in MIMICS, led to the large predicted uncertainty.
We therefore, anticipate substantial uncertainty reduction in these models once more global data
are available to inform the models in the future. On the other hand, the complex model structures
may also contribute to the large uncertainty. In a previous study, Hararuk et al. (2014)¹ reported
small uncertainty in soil C prediction by a conventional model with 20 free parameters. The
number of free parameters are comparable to that in MIMICS (22) and is greater than that in the
vertically-resolved model (13). Therefore, it is likely that the two alternative model structures at
least in part increased the projected uncertainty. In addition, slightly larger uncertainty and fewer
model parameters in the vertically-resolved model than in the MIMICS also support that the
greater uncertainty is likely caused by model structure, if not solely.” (Lines 234-245).

“We simplified the two non-microbial models in terms of their environmental modifiers, soil
water scalar in particular, given data limitation and potential equifinality by the complex
calculation of soil water scalar. This simplification may under-estimate the uncertainty in
predictions by the two models; however, it is less likely for the conventional Century-type model
to reach the similar magnitude of uncertainty in MIMICS even with the full representation of soil
water scalar due to the large difference.” (Lines 307-312).

“We acknowledge that besides transfer coefficients in the two non-microbial models and the
modifiers in the microbial model, data related to other aspects of model processes may be also
critical, but not revealed in our study due to model simplification. For example, data related to
derivation of soil water scalar such as soil water content and water potential may be needed
given that we simplified the calculation of water scalars by directly using the default parameters
in the two non-microbial models and that there is under-representation of soil water impact in
microbial models” in discussion (Lines 256-263).

“Overall, whether the vertically-resolved models and microbial models are better representations
of mechanisms for soil C dynamics remain debatable. However, they represent updated
knowledge and important alternate model structures to enhance confidence in prediction. Our
findings suggest that the scientific community should include alternative model structures in
future ensemble model predictions and comparisons to increase projection confidence.” (Lines
329-334).

Please see our detailed responses below to each specific comment.

[Comment 3] Although I don't disagree with this conclusion “reducing the uncertainty of
microbial models requires more observations than are available at the present time”, I'm still
concerned the analysis is cast too narrowly to honestly make the claim. For example, it's really
the water scalar in CLM4.5 that shuts down decomposition in permafrost systems. I wonder if
the low temperature sensitivity and large permafrost diffusion term (D2) reflects this omission?
I'd suspect that if the authors really considered temperature vs. moisture sensitivities in the

conventional models they also would have wider posterior estimates (fig 2) and future
projections (Fig. 4).

**[Response]** We thank Dr. Wieder for such thoughtful comments. We agree that model
simplification, such as not involving soil water scalar in the data assimilation process in the two
non-microbial models, could under-estimate the uncertainty in model projection. Considering
water scalar estimation in the conventional model may cause wider prediction. However, it is
less likely to reach the similar magnitude of uncertainty in the vertically-resolved model and
MIMICS due to the large difference (Lines 307-312).

Specifically, we agree with Dr. Wieder that in addition to temperature scalar and depth scalar,
water scalar is another key parameter controlling soil C decay, especially in permafrost soils. We
need to mention here that the water scalar is included in our algorithm. We used default
parameter values in CLM 4.5 instead of treating those relevant parameters as free parameters for
constraint for three reasons:

1) In contrast to the direct and simple calculations of temperature and depth scalar, calculation of
soil water scalar (r_w) is complex and comprised of multiple steps in CLM 4.5 including:

$r_w = \frac{\log(\frac{\Psi_{min}}{\Psi})}{\log(\frac{\Psi_{min}}{\Psi_{max}})}$, where $\Psi = \Psi_{sat} \times S^{-B}$ where Ψ is soil water potential, Ψ_{sat} is saturated soil

water potential, S is soil wetness derived using $S = \frac{\theta_{liq}}{\theta_{sat} - \theta_{ice}}$, where θ_{liq} is volumetric water

content, θ_{sat} is saturated volumetric water content, and θ_{ice} is volumetric water content

contributed by ice; B is Clapp and Hornberger parameter. In addition, Ψ_{sat} depends on soil

texture. Limited by data, this complex calculation of soil water scalar itself could generate much
equifinality issue.

2) To be consistent in model parameters among these three models: the microbial model,
MIMICS does not simulate effect of soil water.

3) The barely change (around 27%) in soil water content under the future climate change
indicates limited impact of it on soil C projection.

Please note that even though we did not include soil water scalar in our model parameter
estimation, we have this process (water limitation) in our algorithm with default parameter
values in CLM 4.5. Therefore, the low temperature sensitivity may be due to the fact that change
in soil temperature differs along the added soil layers instead of soil water scalar; and large
diffusion term D_2 reflected the assumption that cryoturbation in permafrost soils is larger than
bioturbation in non-permafrost soils.

The issue Dr. Wieder pointed out here arise to a more general question: whether the larger
uncertainties in prediction by the vertically-resolved model and MIMICS are engendered by high
degrees of model freedom or model structures or both? It is beyond the scope of our current
study, but we discussed this question in the revised manuscript: “The large uncertainty in the
vertically-resolved model and MIMICS might be engendered by either high degrees of model
freedom or complex model structures, or both. Many unconstrained parameters due to data
limitation, especially in MIMICS, led to the large predicted uncertainty. We therefore, anticipate
substantial uncertainty reduction in these models once more global data are available to inform
the models in the future. On the other hand, the complex model structures may also contribute to
the large uncertainty. In a previous study, Hararuk et al. (2014)¹ reported small uncertainty in
soil C prediction by a conventional model with 20 free parameters. The number of free
parameters are comparable to that in MIMICS (22) and is greater than that in the vertically-
resolved model (13). Therefore, it is likely that the two alternative model structures at least in
part increased the projected uncertainty. In addition, slightly larger uncertainty and fewer model
parameters in the vertically-resolved model than in the MIMICS also support that the greater
uncertainty is likely caused by model structure, if not solely.” (Lines 234-245).

[Comment 4] Similarly, by omitting parameter estimate related to respiration coefficients (or
MGE) it seems as though that another equifinality issues is conveniently sidestepped related to
the K and F matrix in the first order models. Again, this generates several concerns related to the
research priorities identified in the discussion / conclusion that focus on transfer coefficient and
turnover rates (i.e., that are only related to temperature sensitivities in the present study).

[Response] Actually we did not omit respiration coefficients in the two non-microbial models.
They were coded in our algorithm and they equal to $(1-f_{i,j})$. Relevant information has been added
(Lines 353-354).

We have toned down the conclusion about focusing on transfer coefficient and turnover rates by
adding “We acknowledge that besides transfer coefficients in the two non-microbial models and
the modifiers in the microbial model, data related to other aspects of model processes may be
also critical, but not revealed in our study due to model simplification. For example, data related
to derivation of soil water scalar such as soil water content and water potential may be needed
given that we simplified the calculation of water scalars by directly using the default parameters
in the two non-microbial models and that there is under-representation of soil water impact in the
microbial model”. (Lines 256-263)

However, we did not consider MGE as a free parameter to constrain due to the fact that we used
microbial biomass data as an input to the MIMICS model while calculating steady state. As a
result, MGE is not involved in calculating steady state soil C in MIMICS. However, we did use
the default MGE values in MIMICS for further prediction. Please see equation 3.4 and 3.5 in
Methods.

[Comment 5] I understand this reductionist approach is necessary to numerically constrain
models. Given data limitations, however, are we at risk of being overly confident in getting the
wrong answers for any multitude of the wrong reasons because of these simplifying
assumptions?

Collectively, I guess these amount to philosophical questions, not particular scientific critiques of
the manuscript. Perhaps a few lines can be added to the discussion along these lines, but this is a
broader philosophical issue we can debate for years to come!

[Response] We really appreciate these thought-provoking comments by Dr. Wieder.

First, we would like to echo Dr. Wieder’s statement that these lines of issues here amount to
philosophical questions which the research community is faced with as a whole. We appreciate

this opportunity to reinforce that there are potential limitations in the model development for
obtaining the ‘truth’ in a target model domain with multiple wrong reasons or mechanisms. For
example, atmospheric CO₂ enrichment may promote plant growth and enhanced plant growth in
turn may accelerate soil C decay through priming effect. This phenomenon can be simulated
using a one-pool soil C decay model with changing turnover rate to represent priming effect²;
however, by parameterizing a two-pool model, the same phenomenon can be captured without
changing the turnover rates³. In another example, most microbial models only consider
temperature effect on microbial activity, due mostly to the lack of knowledge on whether soil
moisture constrains microbial activity primarily through desiccation or diffusion of substrate⁴.
The missing mechanism from water constraint could engender the right simulation results with
low confidence. Therefore, the issue that models obtain the ‘truth’ with wrong reasons, is one of
the ultimate questions in modeling community.

Before we find the best way out of the challenge, it would be reasonable for the whole scientific
community to keep updating the current models with new findings, theories, mechanisms and
data. By doing so, do we increase confidence in model prediction. And it is the logic path for
advance in model development in biogeochemistry or any other scientific discipline. Meantime,
it would be great if we honestly discuss our findings and how these limitations affect our
conclusions. We think that it is very necessary to convey this information to the community. We
added the information in the text. “Overall, whether the vertically-resolved models and microbial
models are better representations of mechanisms for soil C dynamics remain debatable.
However, they represent updated knowledge and important alternate model structures to enhance
confidence in prediction. Our findings suggest that the scientific community should include
alternative model structures in future ensemble model predictions and comparisons to increase
projection confidence.” Please see Lines 329-334.

We therefore took Dr. Wieder’s suggestions and have carefully interpreted and discussed our key
findings in the revised manuscript in multiple places (for example, Lines 234-245, Lines 256-
263, Lines 307-3012 Lines 329-334). Please also refer to response to comment 2 for more
information.

Technical concerns:

[Comment 6] Line 55, I don't think the Todd-Brown papers look at 16 models

**[Response]** Todd-Brown et al. ⁵ looked at 16 models from 11 model centers. Please see Table 1
in Todd-Brown et al. However, due to that there is high similarity between the models from the
same center and in some cases the models from the same center only differ in modeled scenarios,
researchers often cite it as 11 models. We decided to change to 11 models, too.

[Comment 7] Line 105, please include posterior estimates in tables 1-3, as suggested in the text

**[Response]** Mean and standard error have been added.

[Comment 8] Line 115, would the high D2 value be analogous to cryoturbation in permafrost
soils?

**[Response]** 'D' represents diffusivity as explained in Koven et al. 2013⁶, "*diffusive transport*
*occurring as a result of mixing by biological or physical processes*". Therefore, D₂ would not be
completely analogous to cryoturbation, but includes both cryoturbation and biotic mixing by
such as movements of soil animals (bioturbation). However, D₂ is mostly contributed by
cryoturbation in permafrost soils.

[Comment 9] Line 119. Several aspects of these findings are striking.

• The diffusion term (D₂) seems critical for increasing permafrost C stocks in the vertically
resolved model, and seems to largely be responsible for the correction of high latitude biases in
the single layer model (Fig. 3). Isn't this just a fudge factor that allows for spatial 'corrections' in
permafrost regions to better match the permafrost (NCSCD) data?

**[Response]** D₂ is not a fudge factor. It carries ecological and physical meaning in the vertically-
resolved model. D₂ is a specific diffusion term for permafrost soils. D₂ differs from D₁ (diffusion
for non-permafrost soils) in the perspective that D₂ is mainly contributed by cryoturbation and D₁
is mainly contributed by bioturbation. Model assumes that faster diffusion of permafrost soils
than non-permafrost soils due mainly to the cryoturbation, and the assumption is consistent with
field measurements⁶.

[Comment 10] • The low Q10 values suggest that larger stocks are being stored at depth, and in

slow/passive pools, which would have a lower climate sensitivity to climate change given their
inherently lower decay rates. This may ultimately be true, but several recent paper points to the
potential vulnerability of permafrost C (e.g. Comane et al. 2017; Koven et al. 2017). * Note I
realize the Koven paper is just out, but it suggests a yet lower e-folding depth is likely more
appropriate for the same configuration of CLM4.5 and presents a quasi-independent dataset that
can be used to evaluate SOM models.

**[Response]** We agree with Dr. Wieder that low Q_{10} values may lead to large C stock in deeper
and slow/passive soil pools. However, Q_{10} is not the only dominant factor determining the
emergent climatological Q_{10} pattern. We generated the relationship between inferred turnover
rate and mean air temperature with a Q_{10} value of 1.02 in our constrained CLM4.5, following the
procedures in Koven et al., 2017⁷ and Wieder et al., 2017⁸ (Figure R1). We also see high
climatological Q_{10} in cold regions and gradual decrease in climatological Q_{10} with increased
temperature.

The lower e-folding depth in Koven et al.⁷ suggests lack of depth impact on decomposition
(“same property between surface soil C and sub-surface soil C”). I wonder if we can change
other parameter values such as Q_{10} , decay rates or add more processes to generate the same
emergent behavior. It would be a very interesting question to explore. With our data assimilation
technique, we can continue with this research line by adding one more constraint (i.e., the
emergent relationship in Koven et al.⁷) to our algorithm. Then we would reveal whether it is the
model structure or parameter values that lead to the emergent relationship. Actually, the fact that
another conventional model parameterization without depth (CASA-CNP) showed improvement
on the emergent relationship⁸ indicates that there are likely multiple ways to generate the
emergent relationship.

**Figure R1** The emergent relationship between mean air temperature and inferred turnover rate in
CLM 4.5 with Q_{10} value of 1.02.

[Comment 11] • CLM4.5 was tuned up using the vertically resolved isotope data discussed in the
paper (Koven et al. 2013). Do these fitted global parameter combinations maintain soil profiles
that look anything like the observations Charlie used in his paper? Maybe this is beyond the
scope of the paper, but do the spatial distributions of stocks in the vertically resolved simulation
correspond to observed estimates (Fig. S3)?

**[Response]** It is comparable between observed estimates and modeled results in terms of vertical
distribution (Figure R2). We actually have another manuscript in progress relevant to evaluating
the uncertainty in the vertically-resolved model simulation by comparing model results with soil
carbon profile in multiple sites over the globe.

**Figure R2** Normal distribution of modeled soil carbon content in the vertically-resolved model
 at different depth.

[Comment 12] Line 127, Fig S7. Are these total values for CLM4.5 for the top 100 cm soil, or
 the whole 3.5 m profile?

**[Response]** Top 100 cm for the non-permafrost soil and down to 3 meters for the permafrost soil
 for CLM4.5. We present this data to make it comparable to the observations. Relevant
 information has been added to the figure legends (Fig S7 in revised manuscript).

[Comment 13] Line 162, I'm still unclear how or why the authors prescribed initial conditions
 (Fig. 4). I thought the purpose of the data assimilation that provides the bulk of the paper was to
 generate initial conditions that reflect observationally derived estimates? By proscribing initial
 conditions won't there be spurious trends in the transient soil C response?

**[Response]** We meant to show how change in initial condition affect model projection. We agree
 with Dr. Wieder and reviewer #2 that presenting this does not provide much more novel
 information, but confusion. Therefore, we have deleted results related to the prescribed initial
 condition. The changes are reflected in multiple places (such as in title and Figure 4).

[Comment 14] Line 267, parametric and structural (process) uncertainty strike me as distinct
issues. Yes, adding processes also increases parameters, but I feel distinguishing among these
sources of uncertainty is important

**[Response]** Yes. Totally agree. We have deleted the sentence to avoid confusion.

[Comment 15] Line 286 Part of the assumption in MIMICS is that environmental change (e.g.
increased plant productivity) will increase microbial biomass pools (Wieder et al. 2015). Is
microbial biomass allowed to fluctuate in the transient simulations, or is it held constant in the
runs? Please clarify here (and? in the methods?).

**[Response]** Yes. Microbial biomass is allowed to fluctuate in the transient simulations. Relevant
information has been added in the methods (Lines 422-431).

[Comment 16] Line 313, is this statement still true with the new analysis using the HWSO?

**[Response]** Relevant information has been modified with the new results (Lines 323-334).

[Comment 17] On an unrelated note, I wonder if the various parameter estimations actually
afford enough SOM turnover to generate enough inorganic N to meet plant and heterotrophic N
demand in the first-order models. The present approach assumes that modifications to the
parameter estimates don't influence the nitrogen scalar applied in CLM4.5 (line 339). This is
impossible to assess without looking at litter dynamics too (which generally immobilize N), but
it's an interesting feedback that's not considered in the present analysis. I'm not suggesting it
should be included in the analysis but was struck by this limitation of the approach (or
opportunity to throw out parameter combinations that may make the N cycle unrealistic).

**[Response]** We completely agree with Dr. Wieder's comments on feedback of inorganic N to
soil C decay. We acknowledge this limitation along with another feedback from water cycle.
However, it is beyond the scope of our current study and not likely to change our conclusion
with these feedbacks. Moreover, it is very data-demanding to investigate these feedbacks. We are
exploring along this research line at site levels with vast amount of data⁹.

[Comment 18] Just to clarify, CLMcn has a distinct structure and parameterization used in

version 4 of the Community Land Model that includes 3 litter and 4 soil carbon pools. This
structure is not applied here and should not be referred to in the manuscript (I think it's currently
only used in the response to reviewers).

**[Response]** Thank you. They have been updated in the manuscript. We now call it the
conventional Century-type model.

Methods & Display items:

[Comment 19] Line 343. Why are environmental scalars being averaged over the 10 soil layers
in CLM4.5? This covers a depth interval of 0-3.5 m, whereas soil C pools are only estimated for
0-1 m in HWSD and in MIMICS? (see also comments above related to the C stocks for vertically
resolved model).

**[Response]** We apologize for this confusion. We modeled soil C down to 1 meter
in non-permafrost soils in CLM 4.5 with HWSD, and modeled soil C down to 3 meters in
permafrost soils with NCSCD (Lines 524-528). We still used the whole 10 soil layers in CLM
4.5 due to the fact that soil C in deeper layers can feedback to shallower layers through vertical
mixing. However, as we mentioned, we did not compare the soil C of the whole 10 layers to the
database.

In order to keep the environmental scalar consistent among the three models, we therefore
averaged the whole 10 soil layers for the conventional model and MIMICS.

However, we noticed that we predicted the soil C down to 3 meters in MIMICS, which is not
consistent with model assumption (1-meter soil depth in MIMICS). The 3-meter simulation blew
up the uncertainty to be unreasonable (as also pointed out by Reviewer #2). To be consistent
with the model assumption in MIMICS¹⁰, we therefore predicted the soil C down to 1 meters in
the revised manuscript. Now the uncertainty dropped by a substantial amount, larger than the
conventional model, but slightly less than the vertically-resolved model. The result looks more
reasonable and does not change our conclusion (model structures amplify predicted uncertainty).

[Comment 20] Line 524. I'm still not really clear on nuances of the future scenarios work. How
did the authors transition from this data atmosphere to the modeled (RCP8.5 atmosphere)? Is it

worth showing (or describing) changes in litter inputs, temperature, moisture, and their spatial
distribution associated with these simulations in the SI? There are often abrupt step changes
when moving from a data to model atmosphere. I'd assume such changes would cause more non-
linearities in MIMICS. Thus, to what degree is the uncertainty communicated in Fig. 4 related to
the switch between atmospheric forgings?

**[Response]** The variables needed to drive the three models for future projection are
environmental scalars and carbon input to soil. They are outputs from running the NCAR full
model CLM 4.5 with atmospheric forcing under RCP 8.5 from CESM. Spatial distribution of the
changes over the projection periods has been updated in supplementary materials. Based on both
spatial and temporal changes, we did not observe obviously abrupt change in litter input, soil
temperature and soil water content (Figure S10 and S11 in the revised manuscript).

[Comment 21] Line 533, as above, why 10 soil layers?

**[Response]** Please see our response to comment 19.

[Comment 22] Can another column be added to Tables 1-3 showing the mean (or median)
estimates for posterior values for each parameter (and their uncertainty)? I like the information
communicated in Fig. 2, but this additional data in the appropriate tables would be useful too.

**[Response]** Mean and standard error have been added according to the comment.

[Comment 23] Is vertical mixing (Fig 1b) the same as diffusivity (Table 2)? If so, can consistent
terminology be used throughout on display items

**[Response]** Yes. We consistently call it diffusivity throughout the text according to Koven et al.
2013⁶.

[Comment 24] Fig 1c: I'm still confused where the C fluxes go here. If microbial biomass is
fixed based on the Xu dataset (line 443), but C inputs enter the MICr and MICK pools is the
model effectively being tuned to get the 'right' microbial biomass pools? Without MGE, the only
modifier on microbial biomass is the tau term, which is also critical for determining steady state
SOM pools (Wang et al 2013)? To maintain these biomass stocks in the RCP scenario, tau would

have to increase, and would necessarily build more SOM pools (without a feedback to increase
decay rates via larger microbial biomass pools)? Am I understanding this correctly?

**[Response]** We apologize for the confusion. First, microbial biomass is not fixed in the
projection (please also see response to comment 15); second, the MGE for SOMa absorption is
used in our model (equation 3.4 and 3.5) with the prescribed values in Wieder et al., 2015.
According to the conceptual diagram (Fig. 1), C input is the same as the input in the other two
non-microbial models. The input to SOMp and SOMc follow the equations (A15 and A16) in
Wieder et al. (2015): input to SOMp, $R_{l-p} = f_m \times \text{total_input}$; input to SOMc, $R_{l-c} = f_s \times$
total_input ; the rest of the input goes to soil microbes modified by their uptake rates.
Specifically, R_{l-r} and R_{l-k} are the input to r- and k-selection soil microbes ($R_{l-r} = (U_{m-r} + U_{s-r}) / (U_{m-r} + U_{m-k} + U_{s-r} + U_{s-k}) \times (\text{total_input} - R_{l-p} - R_{l-c})$;
$R_{l-k} = (U_{m-k} + U_{s-k}) / (U_{m-r} + U_{m-k} + U_{s-r} + U_{s-k}) \times$
$(\text{total_input} - R_{l-p} - R_{l-c})$). U_{m-r} and U_{m-k} are the uptakes of metabolic litter by r- and k-selection
microbes, respectively; and U_{s-r} and U_{s-k} are the uptakes of structural litter by r- and k-selection
microbes, respectively; all the U's are calculated with default parameters in MIMICS and litter
stock from CLM 4.5 with the sole purpose of normalizing the input to r- and k-selection
microbes. Please note that there are no free parameters involved in the calculation. All the input
(R's) add up, equal to the total input, which is the same as the other two model input. Relevant
information has been added in the revised manuscript (Lines 422-431).

**[Comment 25]** Table 3. I'm not sure what the following parameter refer to in the modified
implementation of MIMICS, as litter stocks have been removed and chemically protected SOM
doesn't flow into microbial biomass (Fig. 1) (V_{mrc} , V_{mkc} , K_{mrc} , K_{mkc}). As such, I wonder if
these parameters should be removed from the analysis, or at least their description in the table
should be clarified? Also, how are K_{mrc} , K_{mkc} different from K_{Or} and K_{Ok} ?

**[Response]** Sorry for the confusion. Litter stocks are indeed removed, but these are modifiers for
the uptake process from chemically protected SOM to available SOM (equation 1 and 2 in the
main text in Wieder et al., 2015). K_{mrc} and K_{mkc} are modifiers for half saturation constant
(K_m) in equation 2 in main text (equation 2 in Wieder et al., 2015) and K_{Or} and K_{Ok} are further
modifiers for K_m in equation A10 (in Wieder et al., 2015).

[Comment 26] MIMICS makes explicit assumptions about the kinetics and turnover of r vs. K
microbial communities (Wieder et al. refs in the paper). Do the parameter combinations used in
the data assimilation maintain these relative differences (e.g. faster kinetics and turnover of
copiotrophic communities)? If not, then differences between r and K physiology are
meaningless, and distinctions made between their physiology and relative abundance should be
removed from the analysis.

**[Response]** It is not a criterion in our algorithm. However, our result supports the assumption
that r-selection microbial community has faster turnover than the K-selection microbial
community. Please see figure 2c (mean and median of T_r and T_k).

[Comment 27] The colors used in the left panels of Fig 3 are ugly. The RMSE colors are nice,
but they don't communicate the sign of the biases like the left panels do.

**[Response]** The colors have been changed. Hopefully, they are good-looking now:). The signs
on left panes have shown the signs of the biases.

[Comment 28] Fig 5, the whole text is organized with the single layer, multi-layer, and MIMICS
model organization, should this figure follow the same layout (top to bottom)?

**[Response]** Thank you. The figure has been updated.

[Comment 29] Fig 5, what is S_i (right side of figure). It's never mentioned in tables 1-3 (as stated
in the caption), but has large leverage in the first order models?

**[Response]** S_i is initial condition of soil carbon and was explained in the figure caption.

[Comment 30] Fig. S3, doesn't the HWSD just cover 0-100 cm? Please clarify what's shown in
these histograms

**[Response]** Sorry for the confusion. What we showed in Figure S3 is the composite database of
HWSD and NCSCD. The database is comprised by HWSD (down to 1 meter deep) in non-
permafrost and NCSCD (down to 3 meters deep) in the permafrost. We used this composite
database to conduct data assimilation (Lines 485-491 and Lines 524-528 in Methods). More
details have been updated in the figure caption.

Commane R, Lindaas J, Benmergui J et al. (2017) Proc Natl Acad Sci U S A, 114, 5361-5366,
doi:10.1073/pnas.1618567114.

Koven CD, Hugelius G, Lawrence DM, Wieder WR (2017) Nature Clim. Change, advance
online publication doi:10.1038/nclimate3421.

Sincerely,

Will Wieder

Reviewer #2 (Remarks to the Author):

Dear authors

[Comment 31] The uncertainty of the representation of soil carbon dynamics in Earth System
Models is an important issue and this manuscript addresses the structural uncertainty with three
models of increasing complexity.

I have carefully read your replies to previous reviews and find that this revised manuscript has
addressed most of the concerns previously raised.

**[Response]** We thank reviewer #2 for the positive evaluation.

[Comment 32] I am, however, concerned about the spread of the uncertainty in projections made
with the MIMICS model. While I appreciate that this study presents results from offline models,
an uptake of 2,000 Pg C would mean a doubling soil stocks over the course of one century... This
should have been addressed as a main limitation of the study in the paragraph l. 280 onwards.

**[Response]** We thank reviewer #2 for the great comment. The uncertainty in MIMICS was
mainly due to the fact that we used 3-meter soil depth instead of 1 meter. It blew up the

uncertainty. The new results (figure 4 with assimilated data and projection down to 1 meter)
showed reasonable gain or loss and did not change our conclusion.

[Comment 33] Also, the statement that "Our results imply that the scientific community should
include microbial models in future ensemble model predictions and comparisons to increase
projection confidence" contradicts the results. It feels that MIMICS is actually over-
parameterized for the low amount of constraints used in the current study, something that is
referred to as "model flexibility" (1 244).

[Response] We apologize for the confusion. We were trying to convey that microbial model is
an important alternative to the conventional model given that it generates diverse trajectory and
higher precision of spatial distribution of soil C. We further stated in the revised manuscript
"Overall, whether the vertically-resolved models and microbial models are better representations
of mechanisms for soil C dynamics remain debatable. However, they represent updated
knowledge and important alternate model structures to enhance confidence in prediction. Our
findings strongly suggest that the scientific community should include alternative model
structures in future ensemble model predictions and comparisons to increase projection
confidence". (Lines 329-334).

We also discussed that the large uncertainty is not solely caused by over-parametrization, but
together with model structure. "The large uncertainty in the vertically-resolved model and
MIMICS might be engendered by either high degrees of model freedom or complex model
structures, or both. Many unconstrained parameters due to data limitation, especially in
MIMICS, led to the large predicted uncertainty. We therefore, anticipate substantial uncertainty
reduction in these models once more global data are available to inform the models in the future.
On the other hand, the complex model structures may also contribute to the large uncertainty. In
a previous study, Hararuk et al. (2014)¹ reported small uncertainty in prediction by a
conventional model with 20 free parameters. The number of free parameters are comparable to
that in MIMICS (22) and is greater than that in the vertically-resolved model (13). Therefore, it
is likely that the two alternative model structures at least in part increased the projected
uncertainty. In addition, slightly larger uncertainty and fewer model parameters in the vertically-
resolved model than in the MIMICS also support that the greater uncertainty is likely caused by

model structure, if not solely.”. (Lines 234-245).

[Comment 34] These points of discussion are raised in sections starting on l 249 and l 280.

Therefore, I don't think these points have to be addressed with new experiments for this paper,

but perhaps more emphasis should be put on how realistic these simulations are, in the

discussion.

[Response] Please see our response to comment 32.

[Comment 35] I am not sure of the usefulness of the experiments based on "data-derived" initial

conditions. These results are only mentioned once in the text (l 167) and, as the authors correctly

mention, findings are anticipated as the data-derived initial conditions are most likely not the

same as the models' own steady-state. Perhaps you could consider removing this experiment for

the sake of simplicity.

[Response] We agree with the reviewer. We have removed this part in the revised manuscript

and changed multiple places in the title and text to reflect the removal (title and figure 4 in

particular).

[Comment 36] Finally, it seems unrealistic for me to reach exactly 1 when calculating the G-R

criterion across multiple chains. The number of chains used in the MCMC is not indicated,

neither is the precision of the G-R criterion reported in Tables 1, 2 and 3. This needs clarification

as, at the moment, it seems that only one chain has been used which would involve a risk of

being trapped in a local optimum.

[Response] We apologize for the confusion. There are five chains (Line 570, K=5 in the revised

version) used in the calculation. We updated the precision in Table 1, 2, and 3. The codes and

data for the calculation can be found online (<https://github.com/zshi0609/Global-DA-Project>).

Hereafter are some minor suggestions for improvement:

[Comment 37] l 43: "can uptake about 1/3 fossil-fuel CO2 emissions",

[Response] Corrected.

[Comment 38] | 105-106: here or under methods, please indicate how you use the constraint with
the multi-layer model... is it summing up all C stocks in each layer? or do you assimilate data for
0-100, 100-200 and 200-300 cm independently?

**[Response]** We assimilate data for 0-100, 100-200 and 200-300 cm independently. Information
has been added in the text (Lines 524-528).

[Comment 39] | 153-157: please give a quantitative comparison of the uncertainty between the
different model structures, e.g. how much larger is the uncertainty in the vertically-resolved /
microbial model compared to the conventional model?

**[Response]** Quantitative comparisons have been updated in the text (Lines 156-158).

[Comment 40] | 157-160: see my main point, how realistic is it for global soils to uptake 2,000
585 Pg C over the course of a century?

**[Response]** Please see our response to comment 32.

[Comment 41] | 221-222: what about other models used in e.g. cmip5? I assume that our results
fall well within the uncertainty of projections obtained by other conventional soil carbon models
used in most ESMs.

**[Response]** Our results indeed fall within the uncertainty of the CMIP 5 models⁵.

[Comment 42] | 305: I do not think the term C-climate feedback is adapted here. All experiments
are made with prescribed atmospheric conditions. What your results indicate is a gain or loss of
soil C.

**[Response]** What we are trying to convey here is that the consistent loss of soil C to atmosphere
due to climate change provides positive feedback to climate system. We think that we are on the
same page with the reviewer.

[Comment 43] | 348-350: what is the thickness of each layer?

**[Response]** The thickness of each layer has been added in the figure (Figure 1).

[Comment 44] l 468-472: these two sentences are redundant

[Response] Thank you. We have deleted the second sentence in the revised version.

[Comment 45] l 527: where do the atmospheric data used to drive CLM come from?

[Response] We used the atmospheric data from the NCAR Earth System Model, Community Earth System Model (CESM) output for the Representative Concentration Pathway 8.5 (RCP8.5) experiment. Relevant information has been added (Lines 550-555).

[Comment 46] Table 1: what is the precision of the G-R statistic reported here? It seems unrealistic to reach exactly 1 when calculating it across several chains. The number of chains is also not reported.

[Response] There are five chains (Line 570, K=5 in the revised version) used in the calculation. We updated the precision in Table 1, 2, and 3. The codes and data for the calculation can be found online (<https://github.com/zshi0609/Global-DA-Project>).

[Comment 47] fig 2: many parameters in MIMICS are not constrained, is it a problem?

[Response] The lack of constraint on these parameters suggests that insensitivity of these parameters to soil C observations. One of the objectives of our study was to show that over-parametrization and complex model structure may lead to equifinality and eventually lead to large uncertainty in projection (Lines 234-245).

[Comment 48] fig S3: please indicate that data for 100-200 and 200-300 cm layers is only available in NCSCD

[Response] Done.

[Comment 49] fig S7: please add the uncertainty of the composite database on the figure

[Response] Great suggestion. Uncertainty has been added.

[Comment 50] fig S8: please indicate that the correlation is calculated across the ensemble of
1,000 global simulations

[Response] Added as suggested.

[Comment 51] fig S9: same as above

[Response] Added as suggested.

Literature cited in the response letter

- 1. Hararuk O, Xia J, Luo Y. Evaluation and improvement of a global land model against
soil carbon data using a Bayesian Markov chain Monte Carlo method. *Journal of*
*Geophysical Research: Biogeosciences* **119**, 2013JG002535 (2014).
2. van Groenigen KJ, Qi X, Osenberg CW, Luo Y, Hungate BA. Faster Decomposition
Under Increased Atmospheric CO₂ Limits Soil Carbon Storage. *Science*,
(2014).
3. Georgiou K, Koven CD, Riley WJ, Torn MS. Toward improved model structures for
analyzing priming: potential pitfalls of using bulk turnover time. *Global Change Biology*
**21**, 4298-4302 (2015).
4. Wieder WR, *et al.* Explicitly representing soil microbial processes in Earth system
models. *Global Biogeochemical Cycles* **29**, 1782-1800 (2015).
5. Todd-Brown KEO, *et al.* Causes of variation in soil carbon simulations from CMIP5
Earth system models and comparison with observations. *Biogeosciences* **10**, 1717-1736
(2013).
6. Koven CD, *et al.* The effect of vertically resolved soil biogeochemistry and alternate soil
C and N models on C dynamics of CLM4. *Biogeosciences* **10**, 7109-7131 (2013).
7. Koven CD, Hugelius G, Lawrence DM, Wieder WR. Higher climatological temperature
sensitivity of soil carbon in cold than warm climates. *Nature Climate Change* **7**, 817
(2017).
8. Wieder WR, Hartman MD, Sulman BN, Wang Y-P, Koven CD, Bonan GB. Carbon cycle
confidence and uncertainty: Exploring variation among soil biogeochemical models.
*Global Change Biology*, DOI: 10.1111/gcb.13979 (2017).

- 9. Shi Z, Yang Y, Zhou X, Weng E, Finzi AC, Luo Y. Inverse analysis of coupled carbon–
nitrogen cycles against multiple datasets at ambient and elevated CO₂. *Journal of Plant*
*Ecology*, (2015).
- 10. Wieder WR, Grandy AS, Kallenbach CM, Taylor PG, Bonan GB. Representing life in the
Earth system with soil microbial functional traits in the MIMICS model. *Geosci Model*
*Dev* **8**, 1789-1808 (2015).

Reviewers' comments:

Reviewer #1 (Remarks to the Author):

I appreciate the revisions made to the manuscript. A few minor corrections stood out on reading the revised draft and are listed below.

Of a more technical concern, both reviewers raised concerns / confusion regarding the soil depth being considered. In this revision, Shi and co-authors still maintain 3.5 m soil pools over permafrost regions (but 1.0 m elsewhere). The arbitrary nature of this decision still strikes me as odd, especially when data to 1 m depth are available from the standard model output and observational datasets.

Line 120, technically the physically protected pool in MIMICS doesn't decay (which implies a CO₂ loss in my mind), but is protected by sorption from microbial decomposition.

Line 132, This discussion about 1 vs. 3.5 m depth is confusing in the main text and not really explained in the methods. From the authors' response letter, it appear as though permafrost SOC targets were for 3.5 m depth, not 1 m as elsewhere? More, the decision to tune the models to somewhat arbitrarily different depth intervals seems odd. Why not just use the 1 m data from the NCSCD and HWSD for consistency, globally? Please correct or clarify as needed.

Some of the text in the paragraph beginning ~ line 135 seems redundant with the text immediately preceding it.

Line 137, errors in coastal areas are not apparent in Fig 3?

Figure 4, A dashed horizontal line showing no changes in stocks would be helpful.

Line 159, I'd challenge the assertion that the more complicated models were made more complex to better match contemporary observations so SOC stocks. Instead the added complexity is intended to represent processes that are handled implicitly or omitted in the simpler model.

Lines 231, I'm not sure results support this more subjective conclusion that results '...caution against inference from more sophisticated models...'. More it directly contradicts the final sentence of the main text. Instead, what about "Results from this study highlight that data constraints may limit the ability of data assimilation to reduce uncertainty in more complicated model structures."

Reviewer #2 (Remarks to the Author):

Dear Authors

Many thanks for the revisions. All my previous concerns have been addressed with detailed replies and appropriate additions to the text.

My only comment is that it may make the text clearer to explicitly refer to CLM4.5 and MIMICS instead of the "two alternative models" (e.g. l. 91, 144, 242, 246, 316, 324, 333).

Regards

**Responses to Comments from Reviewers**

(Manuscript number: NCOMMS-17-20128-B)

**Dear Reviewers:**

We appreciate your insightful comments and suggestions. Below, we address the comments and
questions point-by-point. We have made changes in the text accordingly and highlighted them in
red. The major point is listed as follows:

**The choice of soil carbon data:** The first reviewer is concerned with our choice of using soil
carbon data down to 3 meters in the permafrost regions and down to 1 meter elsewhere. We
argue that in contrast to soils elsewhere, permafrost regions contain a huge amount of carbon
stock in deeper soil, which validates the use of deeper soil carbon data. Originally, in the first
version of the submitted manuscript (NCOMMS-17-20128-T), we used the Harmonized World
Soil Database (HWSD) which has soil carbon stock down to 1 meter in both permafrost regions
and elsewhere. Dr. Wieder and the other reviewer suggested us using the permafrost database for
the permafrost regions, the Northern Circumpolar Soil Carbon Database (i.e., NCSCD which has
soil carbon data down to 3 meters) in the previous comments, which we fully agreed. Both
reviewers expected better model-data fitting with the NCSCD data for the two non-microbial
models, which is true in our revised manuscripts (NCOMMS-17-20128-A and NCOMMS-17-
20128-B). Note also that the conclusion does not change whether the 3-m or 1-m deep soil
carbon in permafrost regions was used. Nonetheless, we have added relevant information to
avoid possible confusion (Lines 524-531).

“For the conventional model, we assimilated the data by aggregating all the soil layers
together which is 0-100cm in non-permafrost regions and 0-300cm permafrost regions; for CLM
4.5, we assimilate data for 0-100cm in HWSD for non-permafrost soils, 0-100, 100-200 and 200-
300cm in NCSCD for permafrost soils, independently; In contrast to soils elsewhere, permafrost
regions contain a huge amount of carbon stock in deeper soil, which validates the use of deeper
soil carbon data. For MIMICS, however, we assimilated the soil C data down to 100 cm only due
to the explicit 1-m depth parameterization²⁰.”

Thank you very much for considering our revised manuscript. We hope you will find our
revision satisfactory.

Sincerely,

Zheng Shi

School of Meteorology

University of Oklahoma, OK, USA

Tel: +1 (405) 325-6519

Email: zheng_shi_ecology@outlook.com

**Response to Reviewer #1**

[Comment 1] I appreciate the revisions made to the manuscript. A few minor corrections stood
out on reading the revised draft and are listed below.

[Response] We thank Reviewer #1 for the positive evaluation and appreciate these constructive
comments. Please see our responses below to each comment.

[Comment 2] Of a more technical concern, both reviewers raised concerns / confusion regarding
the soil depth being considered. In this revision, Shi and co-authors still maintain 3.5 m soil
pools over permafrost regions (but 1.0 m elsewhere). The arbitrary nature of this decision still
strikes me as odd, especially when data to 1 m depth are available from the standard model
output and observational datasets.

[Response] In contrast to soils elsewhere, permafrost regions contain a huge amount of carbon
stock in deeper soil, which validates the use of deeper soil carbon data. Originally in the first
version of the submitted manuscript (NCOMMS-17-20128-T), we only used the Harmonized
World Soil Database (HWSD) which has soil carbon stock down to 1 meter in both permafrost
regions and elsewhere. Dr. Wieder and the other reviewer suggested us using the permafrost
database, Northern Circumpolar Soil Carbon Database (i.e., NCSCD which has soil carbon data
down to 3 meters) in the previous comments, which we fully agreed. Both reviewers expected
better model-data fitting with the NCSCD data for the two non-microbial models, which is true
in our revised manuscripts (NCOMMS-17-20128-A and NCOMMS-17-20128-B). Note also that
the conclusion does not change whether the 3-m or 1-m deep soil carbon was used. Nonetheless,
according to this comment, we have added relevant information to avoid possible confusion
(Lines 524-531).

“For the conventional model, we assimilated the data by aggregating all the soil layers
together which is 0-100cm in non-permafrost regions and 0-300cm permafrost regions; for CLM
4.5, we assimilate data for 0-100cm in HWSD for non-permafrost soils, 0-100, 100-200 and 200-
300cm in NCSCD for permafrost soils, independently; In contrast to soils elsewhere, permafrost
regions contain a huge amount of carbon stock in deeper soil, which validates the use of deeper
soil carbon data. For MIMICS, however, we assimilated the soil C data down to 100 cm only due
to the explicit 1-m depth parameterization²⁰.”

[Comment 3] Line 120, technically the physically protected pool in MIMICS doesn't decay
(which implies a CO₂ loss in my mind), but is protected by sorption from microbial
decomposition.

**[Response]** The word “decay” has been changed to “desorption”. (Lines 120).

[Comment 4] Line 132, This discussion about 1 vs. 3.5 m depth is confusing in the main text and
not really explained in the methods. From the authors' response letter, it appear as though
permafrost SOC targets were for 3.5 m depth, not 1 m as elsewhere? More, the decision to tune
the models to somewhat arbitrarily different depth intervals seems odd. Why not just use the 1 m
data from the NCSCD and HWSO for consistency, globally? Please correct or clarify as needed.

**[Response]** We apologize for the confusion. We indeed used all the 1-m deep soil carbon data
globally to constrain the MIMICS due to the explicit 1-m depth parametrization in MIMICS as
suggested by Dr. Wieder in the previous comments. However, for the other two models
(conventional model and CLM 4.5), we used the 3-m soil carbon data in the permafrost regions
and 1-m soil carbon data elsewhere. Please see our response to comment 2. Relevant information
has been added to avoid the confusion (Lines 524-531).

[Comment 5] Some of the text in the paragraph beginning ~ line 135 seems redundant with the
text immediately preceding it.

**[Response]** The preceding paragraph discusses about spatial pattern of soil carbon in HIGH
latitude (Lines 126-133 in the revised manuscript) and this paragraph (Lines 135-142 in the
revised manuscript) discusses spatial pattern of soil carbon in LOW latitude.

[Comment 6] Line 137, errors in coastal areas are not apparent in Fig 3?

**[Response]** Thank you for the comment. We meant to say near-coastal areas. It has been
corrected (Line 137).

[Comment 7] Figure 4, A dashed horizontal line showing no changes in stocks would be helpful.

**[Response]** The dashed line has been added as suggested.

[Comment 8] Line 159, I'd challenge the assertion that the more complicated models were made
more complex to better match contemporary observations so SOC stocks. Instead the added
complexity is intended to represent processes that are handled implicitly or omitted in the
simpler model.

[Response] According to this comment, we deleted “to provide a better fit to contemporary
observations” (Lines 158-160).

[Comment 9] Lines 231, I'm not sure results support this more subjective conclusion that results
'...caution against inference from more sophisticated models...'. More it directly contradicts the
final sentence of the main text. Instead, what about "Results from this study highlight that data
constraints may limit the ability of data assimilation to reduce uncertainty in more complicated
model structures."

[Response] We really like this sentence suggested by the reviewer and replaced the existing one
in the revised manuscript (Lines 230-232).

**Response to Reviewer #2**

[Comment 10] Many thanks for the revisions. All my previous concerns have been addressed
with detailed replies and appropriate additions to the text.

[Response] We thank Reviewer #2 for the positive evaluation.

[Comment 11] My only comment is that it may make the text clearer to explicitly refer to
CLM4.5 and MIMICS instead of the "two alternative models" (e.g. l. 91, 144, 242, 246, 316,
324, 333).

[Response] We have replaced the “two alternative models” with “CLM4.5 and MIMICS” where
appropriate according to this comment.

REVIEWERS' COMMENTS:

Reviewer #1 (Remarks to the Author):

Reference to the 'alternative models' was not replaced with 'CLM4.5 and MIMICS' as requested by R2 on line 242 or 316.

Otherwise I appreciate efforts by the authors to review this manuscript.

Will

Response Letter

Reviewer #1 (Remarks to the Author):

Reference to the 'alternative models' was not replaced with 'CLM4.5 and MIMICS' as requested by R2 on line 242 or 316.

*Otherwise I appreciate efforts by the authors to review this manuscript.
Will*

Response: changes have been made according to the comment (Lines 254 & 329).